# LEARNING OF POPULATION DYNAMICS: INVERSE OPTIMIZATION MEETS JKO SCHEME

**Mikhail Persiianov**
Applied AI Institute, Moscow, Russia
`persiianov.mi@gmail.com`

**Jiawei Chen**
Applied AI Institute, Moscow, Russia
MIRAI[*], Moscow, Russia

**Petr Mokrov**
Applied AI Institute, Moscow, Russia

**Alexander Tyurin**
AXXX, Russia
Applied AI Institute, Moscow, Russia

**Evgeny Burnaev**
Applied AI Institute, Moscow, Russia
AXXX, Russia

**Alexander Korotin**
Applied AI Institute, Moscow, Russia
AXXX, Russia
`iamalexkorotin@gmail.com`

## ABSTRACT

Learning population dynamics involves recovering the underlying process that governs particle evolution, given evolutionary snapshots of samples at discrete time points. Recent methods frame this as an energy minimization problem in probability space and leverage the celebrated JKO scheme for efficient time discretization. In this work, we introduce `iJKOnet`, an approach that combines the JKO framework with inverse optimization techniques to learn population dynamics. Our method relies on a conventional *end-to-end* adversarial training procedure and does not require restrictive architectural choices, e.g., input-convex neural networks. We establish theoretical guarantees for our methodology and demonstrate improved performance over prior JKO-based methods. Source code:

 `https://github.com/MuXauJl11110/iJKOnet`.

## 1 INTRODUCTION

Modeling population dynamics is a fundamental challenge in many scientific domains, including biology (Schiebinger et al., 2019; Moon et al., 2019), ecology (Ayala et al., 1973), meteorology (Fisher et al., 2009; Sigrist et al., 2015; Verma et al., 2024; Price et al., 2025), transportation flows in urban networks (Medina-Salgado et al., 2022), and epidemiology (Wang et al., 2021; Kosma et al., 2023), among others. The task is to infer the underlying stochastic dynamics of a system – typically modeled by stochastic differential equations (SDEs) – from observed marginal distributions at discrete time points. While this problem has been studied extensively in settings where individual trajectories are available (Krishnan

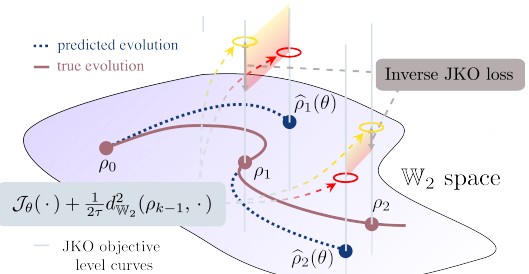

Figure 1: `iJKOnet` working scheme: our method minimizes the **gap** between the optimal values of parametric ($\theta$) JKO functional and suboptimal values obtained at ground truth population measures.

et al., 2017; Li et al., 2020; Brogat-Motte et al., 2024), such data are often unavailable in practice. In many real-world scenarios, we only observe *population-level data*, where it is infeasible to continuously track individual entities. Instead, we are forced to rely on temporally separated and mutually independent snapshots of the population.

---

[*]Moscow Independent Research Institute of Artificial Intelligence

In *single-cell genomics* (Macosko et al., 2015), for example, measuring the state of individual cells typically involves destructive sampling – a process where cells are destroyed during measurement, preventing any further observation of their future behavior. As a result, the data consist only of isolated profiles of cellular populations taken at discrete time points. Reconstructing the continuous developmental trajectories of cells from such fragmented data poses a challenge. A similar situation arises in *financial markets* (Gontis et al., 2010; Yang et al., 2023), where analysts often have access only to marginal distributions of asset prices at specific times, making it necessary to infer the underlying dynamics that govern these distributions. In *crowd dynamics* (Maury et al., 2010; 2011; Wan et al., 2023), modeling the temporal evolution of population densities is crucial for understanding and managing pedestrian flows. In this case, data on individual trajectories are also typically unavailable, and only aggregate distributions at various time points are observed.

A promising approach to address the absence of particle trajectories is the Jordan–Kinderlehrer–Otto (JKO) scheme (Jordan et al., 1998), which models the evolution of a particle system as a sequence of distributions that gradually approach the minimum of a total energy functional while remaining close to the previous distributions. However, implementing the JKO scheme involves solving an optimization problem over the space of probability measures, which is computationally demanding. The first attempt to leverage JKO for learning population dynamics was introduced in (Bunne et al., 2022b, `JKOnet`). While innovative, this approach relies on a complex learning objective and is limited to potential energy functionals – meaning it cannot capture stochasticity in the dynamics. A recent work, (Terpin et al., 2024, `JKOnet*`), proposes replacing the JKO optimization step with its first-order optimality conditions. This relaxation allows modeling more general energy functionals and reduces computational complexity, but it does not support end-to-end training. Instead, it requires precomputing optimal transport couplings (Cuturi, 2013) between subsequent time snapshots, which limits its scalability and generalization.

**Contributions.** In this work, we present a novel approach for recovering the system dynamics based on observed population-level data. Our key contributions are as follows:

1. **Methodology:** In §3.2, we cast the problem of reconstructing energy functionals within the JKO framework as an inverse optimization task. This perspective leads to a novel min-max optimization objective for population dynamics recovery.
2. **Algorithm:** We equip our population dynamics methodology with a conventional end-to-end adversarial learning procedure (§3.3). Importantly, our practical scheme does not pose restrictions on the architectures of the utilized neural networks, which contributes to the scalability.
3. **Theoretical Guarantees:** Under suitable assumptions, we show that our method can accurately recover the underlying energy functional governing the observed population dynamics (§ 3.4).

In §5 and Appendix B, we evaluate our approach on a range of synthetic and real-world datasets, including single-cell genomics. The results show that our method demonstrates improved performance over previous JKO-based approaches for learning population dynamics.

**Notation.** Let $\mathcal{X}$ be a compact convex subset of $\mathbb{R}^D$ equipped with the Euclidean norm $\| \cdot \|_2$. Let $\mathcal{P}(\mathcal{X})$ denote the set of probability measures on $\mathcal{X}$, and let $\mathcal{P}_{ac}(\mathcal{X})$ denote its subset of probability measures absolutely continuous with respect to the Lebesgue measure. For $\rho \in \mathcal{P}_{ac}(\mathcal{X})$, we use $\rho$ to denote both the measure and its density function with respect to the Lebesgue measure. For measures $\mu, \nu \in \mathcal{P}(\mathcal{X})$, we denote the set of couplings (*transportation plans*) between them by $\Pi(\mu, \nu)$. For a measure $\rho \in \mathcal{P}(\mathcal{X})$ and measurable map $T : \mathcal{X} \to \mathcal{X}$, we denote by $T\sharp\rho$ the associated *push-forward* measure, and $\mathrm{id}_{\mathcal{X}} : \mathcal{X} \to \mathcal{X}$ is the identity mapping. $\nabla \cdot F = \sum_{d=1}^{D} \frac{\partial F_d}{\partial x_d}$ denotes the *divergence* operator for a continuously differentiable vector field $F : \mathcal{X} \to \mathcal{X}$.

## 2 BACKGROUND

To describe the evolution of population measures, we use the theoretical framework of *Wasserstein gradient flows* (WGFs). In what follows, we first introduce the preliminary concept of Optimal Transport between probability measures (Villani et al., 2008; Santambrogio, 2015). Then we provide sufficient theoretical background on WGFs; see (Santambrogio, 2017; Figalli & Glaudo, 2023) for an overview and (Ambrosio et al., 2008) for a comprehensive study of WGFs theory. Finally, we get the reader acquainted with the JKO scheme, which is the cornerstone of our approach.

**Optimal Transport.** The (squared) *Wasserstein-2 distance* $d_{\mathbb{W}_2}$ between two probability measures $\mu, \nu \in \mathcal{P}(\mathcal{X})$ is defined as the solution to the *Kantorovich* problem (Kantorovich, 1942):

$$d_{\mathbb{W}_2}^2(\mu, \nu) \overset{\text{def}}{=} \min_{\pi \in \Pi(\mu,\nu)} \int_{\mathcal{X} \times \mathcal{X}} \|x - y\|_2^2 \, d\pi(x,y), \tag{1}$$

where the distribution $\pi^*$ delivering the minimum to (1) is called the *optimal coupling* (or *optimal plan*). If $\mu \in \mathcal{P}_{ac}(\mathcal{X})$, then the Brenier's theorem (Brenier, 1991) establishes equivalence between the Kantorovich formulation (1) and *Monge's* problem (Monge, 1781):

$$d_{\mathbb{W}_2}^2(\mu, \nu) = \min_{T: T\sharp\mu=\nu} \int_{\mathcal{X}} \|x - T(x)\|_2^2 \, d\mu(x), \tag{2}$$

where the optimal transport map $T^*$ is known as the *Monge map*. Moreover, there exists a *unique* (up to an additive constant) convex potential $\psi^* : \mathcal{X} \to \mathbb{R}$ such that $T^* = \nabla\psi^*$ and $(\nabla\psi^*)\sharp\mu = \nu$ (McCann, 1995). In this setting, the optimal coupling in (1) is given by $\pi^* = [\text{id}_{\mathcal{X}}, \nabla\psi^*]\sharp\mu$.

**Wasserstein Gradient Flows.** For clarity, we first consider the concept of gradient flows in Euclidean space $\mathbb{R}^D$ and then move to the space of probability measures $\mathcal{P}(\mathcal{X})$. For Euclidean space $\mathbb{R}^D$, a *gradient flow* is an absolutely continuous curve $x(t)$ starting at $x_0 \in \mathbb{R}^D$ that minimizes an energy functional $J : \mathbb{R}^D \to \mathbb{R}$ "as fast as possible". To find such a curve, one needs to solve an ordinary differential equation (ODE) (Teschl, 2012) of the form $x'(t) = -\nabla J(x(t))$ with initial condition $x(0) = x_0$. The same idea is applicable to the space of probability measures $\mathcal{P}(\mathcal{X})$. If we equip $\mathcal{P}(\mathcal{X})$ with the Wasserstein-2 distance $d_{\mathbb{W}_2}$, we obtain the *Wasserstein space* $\mathbb{W}_2(\mathcal{X}) = (\mathcal{P}(\mathcal{X}), d_{\mathbb{W}_2})$ – a complete and separable metric space (Bogachev & Kolesnikov, 2012). In this case, for an energy functional $\mathcal{J} : \mathcal{P}(\mathcal{X}) \to \mathbb{R}$, the gradient flow in $\mathbb{W}_2(\mathcal{X})$, called the *Wasserstein gradient flow (WGF)*, is an absolutely continuous curve $\rho_t : \mathbb{R}_+ \to \mathcal{P}(\mathcal{X})$ starting at $\rho_0$ that follows the steepest descent direction of $\mathcal{J}$, i.e., solves

$$\partial_t \rho_t = -\nabla_{\mathbb{W}_2} \mathcal{J}(\rho_t), \quad \rho_{(t=0)} = \rho_0, \tag{3}$$

where $\nabla_{\mathbb{W}_2} \mathcal{J}(\rho_t)$ denotes the *Wasserstein gradient* in $\mathbb{W}_2$ given by $\nabla_{\mathbb{W}_2} \mathcal{J}(\rho) = -\nabla \cdot (\rho \nabla \frac{\delta\mathcal{J}}{\delta\rho})$ with $\frac{\delta\mathcal{J}}{\delta\rho}$ denoting the first variation (Chewi et al., 2025) of energy $\mathcal{J}$. Thus, equation (3) can be rewritten in the form of the *continuity equation*, expressing mass conservation under the velocity field $v_t$:

$$\partial_t \rho_t + \nabla \cdot (\rho_t v_t) = 0, \ v_t = -\nabla \frac{\delta\mathcal{J}}{\delta\rho}(\rho_t). \tag{4}$$

There exists an intriguing connection between WGFs and certain partial differential equations (PDEs) (Evans, 2022). In particular, different energy functionals in the Wasserstein space correspond to distinct PDEs (Santambrogio, 2015), some of which are associated with diffusion processes appearing in practice (Gómez-Castro, 2024; Bailo et al., 2024b).

**Examples of PDEs as WGFs.** Consider the *free energy* functional (Carrillo et al., 2003, Eq. (1.3)):

$$\mathcal{J}_{\text{FE}}(\rho) = \underbrace{\int_{\mathcal{X}} V(x) \, d\rho(x)}_{\mathcal{V}(\rho)} + \underbrace{\int_{\mathcal{X} \times \mathcal{X}} W(x-y) \, d\rho(x)d\rho(y)}_{\mathcal{W}(\rho)} + \underbrace{\int_{\mathcal{X}} U(\rho(x)) \, dx}_{\mathcal{U}(\rho)}, \tag{5}$$

where $\mathcal{V}$, $\mathcal{W}$, and $\mathcal{U}$ correspond to the system's *potential*, *interaction*, and *internal* energies, respectively. The PDE corresponding to the WGF driven by this functional is known as the *aggregation-diffusion* equation (ADE). It describes the evolution of density $\rho_t$ under a corresponding velocity field $v_t$ (4). This evolution reflects a balance of three effects: *drift* – driven by the potential $V(x)$, modeling an external field (e.g., gravity, electric potential); *interaction* – governed by the symmetric interaction kernel $W(x-y)$, accounting for non-local effects (e.g., particle interactions, long-range forces); and *diffusion* – modeled by the internal energy $\mathcal{U}(\rho)$, representing the energy associated with the local state of the system (e.g., thermal energy, chemical energy). A *weak* solution to the equation exists if the *no-flux conditions*: $\nabla V \cdot \vec{n} = 0$ on $\partial\mathcal{X}$ holds, where $\vec{n}$ is the outward normal vector on the boundary, and if $W \in \mathcal{C}(\mathcal{X} \times \mathcal{X})$, i.e. $W$ is a continuous function on $\mathcal{X} \times \mathcal{X}$. These conditions ensure that no mass crosses the boundary of $\mathcal{X}$; mass can only be redistributed within $\mathcal{X}$. For a detailed discussion of the existence and classification of solutions, see (Gómez-Castro, 2024).

Such energy functionals are ubiquitous in real-world applications, including physics (Carrillo & Gvalani, 2021), biology (Keller & Segel, 1971; Potts & Painter, 2024; Potts, 2024), economics (Fiaschi & Ricci, 2025), machine learning (Suzuki et al., 2023; Chizat et al., 2024; Nitanda, 2024), and nonlinear optimization (Bailo et al., 2024a), to name a few. For further discussion and references, see the recent surveys (Carrillo et al., 2019a; Gómez-Castro, 2024; Bailo et al., 2024b).

To demonstrate the flexibility of the energy formulation in (5), we examine a few representative cases. When the system's energy is purely internal and given by the negative differential entropy, $\mathcal{U}(\rho) = -\mathcal{H}(\rho) \stackrel{\text{def}}{=} \int \rho(x) \log \rho(x) \mathrm{d}x$, continuity equation (4) reduces to the classical heat equation: $\partial_t \rho = \nabla^2 \rho$ (Vázquez, 2017). Alternatively, when the energy is $\mathcal{J}_{\text{FP}}(\rho) = \mathcal{V}(\rho) - \beta \mathcal{H}(\rho)$, the resulting PDE is the *linear Fokker-Planck equation* with diffusion coefficient $\beta$:

$$\partial_t \rho_t = \nabla \cdot (\nabla V(x) \rho_t) + \beta \nabla^2 \rho_t, \tag{6}$$

A fruitful branch of theoretical and practical research stems from the connection between the Fokker-Planck PDE and stochastic differential equations (SDEs) (Risken & Frank, 1996; Bogachev et al., 2022), the latter describes the stochastic evolution of *particles*. In particular, equation (6) is equivalent (Weinan et al., 2021) to the following *Itô SDE*:

$$\mathrm{d}X_t = -\nabla V(X_t) \, \mathrm{d}t + \sqrt{2\beta} \, \mathrm{d}W_t, \tag{7}$$

where $X = \{X_t\}_{t \geq 0}$ is a $\mathbb{R}^D$-valued stochastic process and $W = \{W_t\}_{t \geq 0}$ is a standard Wiener process (Särkkä & Solin, 2019). In other words, if $X_t \sim \rho_t$ evolves according to (7), then the density $\rho_t$ evolves according to the Fokker-Planck equation (6) in the space of probability measures.

WGFs provide a compelling framework for modeling PDEs and their associated SDEs across various domains, giving rise to a range of methods for approximating solutions to WGFs. These include deep learning-based approaches (Mokrov et al., 2021; Alvarez-Melis et al., 2022; Altekrüger et al., 2023), particle-based methods such as Stein Variational Gradient Descent (SVGD) (Liu & Wang, 2016) and its extensions (Das & Nagaraj, 2023; Tankala et al., 2025), as well as classical discretization techniques in Wasserstein space (Carrillo et al., 2022, see §1.2). Many of these advances build on the celebrated Jordan-Kinderlehrer-Otto (JKO) scheme (Jordan et al., 1998), which we review next.

**The JKO Scheme.** A classical method for solving ODEs in Euclidean space is the *implicit Euler scheme*. For an (Euclidean) gradient flow $x'(t) = -\nabla J(x(t))$ and $\tau > 0$, this scheme approximately solves this ODE by iteratively applying the proximal operator (Parikh et al., 2014, §1.1): $x_{k+1} = \text{prox}_{\tau J}(x_k) = \arg\min_{x \in \mathbb{R}^D} \left\{ J(x) + \frac{1}{2\tau} \|x - x_k\|_2^2 \right\}$. Compared to the explicit Euler scheme, this approach offers improved numerical stability (Butcher, 2016). Jordan, Kinderlehrer, and Otto (Jordan et al., 1998) extended this idea to the space of probability measures, introducing a variational time discretization of the Fokker–Planck equation (6), now known as the *JKO scheme*:

$$\rho_{k+1}^\tau = \arg\min_{\rho \in \mathcal{P}(\mathcal{X})} \left\{ \mathcal{J}(\rho) + \frac{1}{2\tau} d_{\mathbb{W}_2}^2(\rho, \rho_k^\tau) \right\} = \text{JKO}_{\tau \mathcal{J}}(\rho_k^\tau), \quad \rho_0^\tau = \rho_0, \tag{8}$$

where $\tau > 0$ is the time step. As $\tau \to 0$, the sequence $\rho_k^\tau, k \in \mathbb{N}$ converges to the continuous solution $\rho_t$ of the Fokker–Planck equation (6). Later, the JKO scheme's convergence was generalized (Ambrosio et al., 2008, Thm. 4.0.4) for the free energy functional $\mathcal{J}_{\text{FE}}$ (5). Note that since $d_{\mathbb{W}_2}^2(\rho, \rho_k^\tau) \geq 0$, the energy is non-increasing along the sequence: $\mathcal{J}(\rho_{k+1}^\tau) \leq \mathcal{J}(\rho_k^\tau)$. This monotonicity property is often utilized in JKO-based algorithms (Salim et al., 2020) in Wasserstein space.

## 3 IJKONET METHOD

We begin by formally stating the problem addressed in our work (§3.1), then we develop our methodology (§3.2). Finally, we discuss the practical (§3.3) and theoretical aspects (§3.4) of our approach.

### 3.1 PROBLEM STATEMENT

In our work, we address the problem of recovering the underlying energy functional $\mathcal{J}^*$ that governs the evolution of a density $\rho_t \in \mathcal{P}_{ac}(\mathcal{X})$ in (4) based on marginal population measures (Bunne et al., 2022b; Lavenant et al., 2024). Specifically, we are given independent samples from marginals $\{\rho_k\}_{k=0}^K$ of the evolving distribution $\rho_t$ at corresponding time points $t_0 < t_1 < \cdots < t_K$. Importantly, each distribution $\rho_k$ may be represented by a different number of samples. As discussed in §2, the JKO scheme approaches the continuous-time dynamics as the step sizes $\Delta t_k = t_{k+1} - t_k$ are small enough. This motivates our modeling assumption that the ground truth sequence of measures $\{\rho_k\}_{k=0}^K$ follows the scheme $\rho_{k+1} = \text{JKO}_{\Delta t_k \mathcal{J}^*}(\rho_k)$.

Although the time intervals $\Delta t_k$ between observations may vary, corresponding to non-uniform step sizes in the JKO scheme, we assume equal spacing $\Delta t_k = \tau$ in the remaining text for simplicity.

## 3.2 METHODOLOGY AND LOSS DERIVATION

In this section, we assume $\rho_k \in \mathcal{P}_{ac}(\mathcal{X})$ and minimization is always taken over $\rho \in \mathcal{P}_{ac}(\mathcal{X})$ in order to utilize Brenier's theorem. The key idea of our method builds on *inverse* optimization techniques (Chan et al., 2025) and is demonstrated in Figure 1. Thanks to assumption $\rho_{k+1} = \text{JKO}_{\tau\mathcal{J}^*}(\rho_k)$, we can derive an inequality that becomes an equality if a candidate functional $\mathcal{J}$ matches the ground truth functional $\mathcal{J}^*$, since $\rho_{k+1}$ is obtained via the JKO step associated with $\mathcal{J}^*$:

$$\min_{\rho^k} \left\{ \mathcal{J}(\rho^k) + \frac{1}{2\tau} d^2_{\mathbb{W}_2}(\rho_k, \rho^k) \right\} \le \mathcal{J}(\rho_{k+1}) + \frac{1}{2\tau} d^2_{\mathbb{W}_2}(\rho_k, \rho_{k+1}). \quad (9)$$

Moving the right-hand side to the left yields an expression that is always upper-bounded by zero, regardless of the choice of $\mathcal{J}$. Maximizing the resulting gap with respect to $\mathcal{J}$ encourages the candidate functional to approximate the true functional $\mathcal{J}^*$. This yields the following objective:

$$\max_{\mathcal{J}} \sum_{k=0}^{K-1} \left[ \min_{\rho^k} \left\{ \mathcal{J}(\rho^k) + \frac{1}{2\tau} d^2_{\mathbb{W}_2}(\rho_k, \rho^k) \right\} - \mathcal{J}(\rho_{k+1}) - \underbrace{\frac{1}{2\tau} d^2_{\mathbb{W}_2}(\rho_k, \rho_{k+1})}_{\text{independent of } \mathcal{J}, \rho^k} \right] =$$

$$\max_{\mathcal{J}} \sum_{k=0}^{K-1} \min_{\rho^k} \left[ \mathcal{J}(\rho^k) - \mathcal{J}(\rho_{k+1}) + \frac{1}{2\tau} d^2_{\mathbb{W}_2}(\rho_k, \rho) \right] + \text{Const.} \quad (10)$$

Thanks to our assumption $\rho_k \in \mathcal{P}_{ac}(\mathcal{X})$, Brenier's theorem (Brenier, 1991) ensures that each distribution $\rho^k$ appearing in (10) can be written as a pushforward $\rho^k = T^k \sharp \rho_k$ for some transport map $T^k : \mathcal{X} \to \mathcal{X}$. In addition, the Wasserstein distance (2) admits the upper-bound $d^2_{\mathbb{W}_2}(\rho_k, \rho) \le \int_{\mathcal{X}} \|x - T^k(x)\|^2 d\rho_k(x)$. Since the minimizations over $\rho^k$ (or equivalently over $T^k$) are independent across $k$, the sum and the minimization can be interchanged. Applying these observations, we end up with our *final loss objective* equivalent to (10):

$$\max_{\mathcal{J}} \min_{T^k} \mathcal{L}(\mathcal{J}, T^k) \overset{\text{def}}{=} \max_{\mathcal{J}} \min_{T^k} \sum_{k=0}^{K-1} \left[ \mathcal{J}(T^k \sharp \rho_k) - \mathcal{J}(\rho_{k+1}) + \frac{1}{2\tau} \int_{\mathcal{X}} \|x - T^k(x)\|_2^2 \rho_k(x)\,\mathrm{d}x \right]. \quad (11)$$

The key idea is that the inner minimization over maps $T^k$ approximates a JKO step for a given energy functional $\mathcal{J}$; that is, each optimal map $T^k_{\mathcal{J}} = \arg\min_{T^k} \left[ \mathcal{J}(T^k \sharp \rho_k) + \frac{1}{2\tau} \int_{\mathcal{X}} \|x - T^k(x)\|_2^2 \rho_k(x)\,\mathrm{d}x \right]$ pushes $\rho_k$ to the JKO-updated distribution $\hat{\rho}_{k+1} = \text{JKO}_{\tau\mathcal{J}}(\rho_k)$. The outer maximization then drives $\mathcal{J}$ toward the true functional $\mathcal{J}^*$, causing the pushforward distributions $\hat{\rho}_{k+1} = T^k_{\mathcal{J}} \sharp \rho_k$ to approach $\rho_{k+1}$, see illustration in Figure 1.

## 3.3 PRACTICAL ASPECTS: METHOD PARAMETRIZATION AND LEARNING PROCEDURE

We denote by $\theta \in \Theta$ and $\varphi \in \Phi$ the parameters of sufficiently expressive function classes (e.g., neural networks) used to approximate the candidate functional $\mathcal{J}$ and the transport maps $T^k$, respectively. Specifically, $\mathcal{J}_\theta$ and $T^k_\varphi$ are their parameterized counterparts.

**Mapping Parameterization.** Building upon (Benamou et al., 2016), prior works (Mokrov et al., 2021; Alvarez-Melis et al., 2022; Bunne et al., 2022b) typically parameterize the transport maps as $T^k_\varphi = \nabla \psi^k_\varphi$, where $\psi^k_\varphi$ are modeled using input-convex neural networks (ICNNs) (Amos et al., 2017). However, ICNNs suffer from poor scalability in high-dimensional settings (Korotin et al., 2021b). In contrast, since our objective (11) imposes *no convexity constraints*, we parameterize $T^k_\varphi$ directly using standard architectures like MLPs or ResNets (He et al., 2016) This relaxation simplifies optimization and yields improved empirical stability. Alternatively, one can employ more task-specific parametrizations, such as triangular maps (Baptista et al., 2024).

**Energy functional parametrization.** Following (Terpin et al., 2024), we parameterize the energy functional $\mathcal{J}$ as a specific instance of the free energy formulation in (5), where the internal energy term is chosen to be scaled negative differential entropy:

$$\mathcal{J}_\theta(\rho) = \underbrace{\int_{\mathcal{X}} V_{\theta_1}(x)\,\mathrm{d}\rho(x)}_{\mathcal{V}_{\theta_1}(\rho)} + \underbrace{\int_{\mathcal{X} \times \mathcal{X}} W_{\theta_2}(x - y)\,\mathrm{d}\rho(x)\mathrm{d}\rho(y)}_{\mathcal{W}_{\theta_2}(\rho)} + \underbrace{\theta_3 \int_{\mathcal{X}} \log \rho(x)\,\mathrm{d}\rho(x)}_{\mathcal{U}_{\theta_3}(\rho) = -\theta_3 \mathcal{H}(\rho)}. \quad (12)$$

Here $\theta = \{\theta_1, \theta_2, \theta_3\}$ represents the set of learnable parameters: $\theta_1$ and $\theta_2$ are parameters of neural networks that define the potential $V_{\theta_1} : \mathcal{X} \to \mathbb{R}$ and the interaction kernel $W_{\theta_2} : \mathcal{X} \to \mathbb{R}$, respectively; $\theta_3 \in \mathbb{R}$ is a learnable scalar diffusion coefficient, see (7).

**Entropy evaluation.** All the terms in the objective function (11) can be estimated using Monte Carlo integration (Metropolis & Ulam, 1949), except for the internal energy term $\mathcal{U}_{\theta_3}$. To handle this remaining component, we follow (Mokrov et al., 2021, Theorem 1), (Alvarez-Melis et al., 2022, Eq. (11)) and apply the change-of-variables formula, yielding:

$$\mathcal{U}_{\theta_3}(T_\varphi^k \sharp \rho_k) = \mathcal{U}_{\theta_3}(\rho_k) - \theta_3 \int_{\mathcal{X}} \log |\det \nabla_x T_\varphi^k(x)| \, \mathrm{d}\rho_k(x), \tag{13}$$

where $\nabla_x T_\varphi^k$ denotes the Jacobian of the *invertible* transport map $T_\varphi^k$. We note that, during training, the computation of $\log \det$ – even for non-invertible mappings $T_\varphi^k$ – can be effectively handled by modern programming tools; see Appendix C.3 for details. To efficiently compute the gradient of the $\log \det$ term in (13) with respect to $\varphi$, one can use the Hutchinson trace estimator (Hutchinson, 1989), as proposed by (Finlay et al., 2020; Huang et al., 2021). However, in our experimental setting (§5), we find that computing the full Jacobian is tractable and sufficient. For estimating the negative differential entropy $\mathcal{H}(\rho_k)$ in (13), we employ the Kozachenko–Leonenko nearest-neighbor estimator (Kozachenko, 1987; Berrett et al., 2019). Notably, entropy values for the population measures $\{\rho_k\}_{k=0}^K$ can be precomputed prior to training.

**Learning Objective.** To facilitate gradient-based optimization, we employ Monte Carlo estimation to approximate the loss in (11). At each time step $t_k$ (for $k = 0, \ldots, K$), we draw a batch of $B_k$ samples $X_k = \{x_k^1, \ldots, x_k^{B_k}\} \sim \rho_k$ and optimize the following *empirical loss objective*:

$$\widehat{\mathcal{L}}(\theta, \varphi) = \sum_{k=0}^{K-1} \left[ \widehat{\mathcal{J}}_\theta(T_\varphi^k(X_k)) - \widehat{\mathcal{J}}_\theta(X_{k+1}) + \frac{1}{2\tau} \sum_{j=1}^{B_k} \|x_k^j - T_\varphi^k(x_k^j)\|_2^2 \right], \tag{14}$$

where $T_\varphi^k(X_k)$ denotes the batched pushforward used to compute the empirical estimates $\widehat{\mathcal{J}}_\theta$ of the functional $\mathcal{J}_\theta$. We optimize the objective in (14) with respect to $\theta$ and $\varphi$ using a standard gradient descent–ascent scheme; underline{implementation details} are provided in §5 and Appendix C, with the complete training procedure summarized in Algorithm 1 therein.

### 3.4 THEORETICAL ASPECTS

The central idea behind our loss formulation (§3.2) is to minimize the *inverse gap* in (9), which measures the discrepancy between the optimal value (left-hand side) and the expected value (right-hand side) of the JKO step for a candidate functional $\mathcal{J}$. While this objective is intuitively justified, it remains to be shown whether the minimizer $\mathcal{J}_{\min}$ of (11) truly approximates the ground-truth energy functional $\mathcal{J}^*$. Our Theorem 3.1 addresses this question, demonstrating that, under suitable assumptions, optimizing (11) indeed recovers $\mathcal{J}^*$ up to an additive constant that does not affect the dynamics governed by (4). Full details and proofs are provided in Appendix D.

We state our theorem for $K = 1$. When considering $K > 1$, the statement of the theorem holds true independently for each timestep $k = 0, 1, \ldots, K-1$, and the results extend straightforwardly to this more general setting. We further assume that both the ground-truth functional $\mathcal{J}^*$ and the candidate functional $\mathcal{J}$ are purely of potential energy form, i.e., $\mathcal{J}^*(\rho) = \mathcal{J}_{\mathrm{PE}}^*(\rho) = \int_{\mathcal{X}} V^*(x) \, \mathrm{d}\rho(x)$; $\mathcal{J}(\rho) = \mathcal{J}_{\mathrm{PE}}(\rho) = \int_{\mathcal{X}} V(x) \, \mathrm{d}\rho(x)$. For notational simplicity, we denote the loss as $\mathcal{L}(V, T)$ in place of $\mathcal{L}(\mathcal{J}_{\mathrm{PE}}, T)$. For technical purposes, we introduce the modified version of a potential as $V_q := \tau V + \frac{1}{2}\|\cdot\|_2^2 : \mathcal{X} \to \mathbb{R}$; subscript $q$ stands for "quadratic".

**Theorem 3.1** (Quality bounds for recovered potential energy). *Let* $\varepsilon(V) \stackrel{def}{=} \mathcal{L}(V^*, T_{V^*}) - \mathcal{L}(V, T_V)$ *be the gap between the optimal and optimized value of inverse JKO loss* (11) *with internal* $\min_T$ *problem solved exactly, i.e.,* $T_V \stackrel{def}{=} \min_T \mathcal{L}(V, T)$. *Let* $\mathcal{X}$ *be a convex set; (modified) potentials* $V_q$ *be strictly convex and* $\frac{1}{\beta}$-*smooth (see the definition in Appendix D). Then there exists* $C = C(\tau, \beta)$ *such that following inequality holds:*

$$\int_{\mathcal{X}} \|\nabla V^*(y) - \nabla V(y)\|^2 \mathrm{d}\rho_1(y) \leq C\varepsilon(V). \tag{15}$$

Notably, equation (15) compares *gradients* of the recovered and ground truth potentials, hiding the appearance of a redundant additive constant. Importantly, the assumptions of the theorem are not particularly restrictive in practice. Specifically, the smoothness of potentials can be ensured by employing smooth activation functions such as `CELU` (Barron, 2017), `SiLU` (Hendrycks & Gimpel, 2016), `SoftPlus`, and others. Strict convexity, in turn, can be enforced through architectural design choices (Amos et al., 2017). Furthermore, the strict convexity of the (modified) potentials $V_q$ can often be assumed when the step size $\tau$ is sufficiently small. In our experiments (§5), we parameterize $V$ using standard MLPs and observe that this approach performs adequately.

To the best of our knowledge, this work is the first to provide a quality analysis (Theorem 3.1) of JKO-based solvers for population dynamics. At present, our analysis focuses exclusively on the *potential* energy component. Extending this framework to incorporate *interaction* and *internal* energies presents an interesting direction for future research. In contrast, (Wu & Wang, 2025) analyzes the JKO scheme in terms of convergence to the true density of the Fokker-Planck equation (6), accounting for parameter uncertainty in iterative updates. Our work differs by focusing on the ability to restore energy components from data, rather than reconstructing the full density.

## 4 RELATED WORKS

This section reviews research directions most relevant to our work. For an extended discussion of related works, see Appendix A.1. First, we discuss existing methods for modeling the dynamics of $\rho_k$ given a known energy functional $\mathcal{J}$ (§4.1). Next, we focus on approaches closely related to our setting – specifically, learning population dynamics from observed data (§4.2) via the JKO scheme.

### 4.1 SOLVING AGGREGATION-DIFFUSION EQUATION WITH GIVEN ENERGY

This line of research focuses on methods for *solving* aggregation-diffusion equations derived from WGFs with given energy functionals of the form (5). A variety of numerical approaches exist, including mesh-based schemes (Cancès et al., 2023; Jüngel et al., 2024), particle methods (Craig & Bertozzi, 2016; Campos Pinto et al., 2018; Carrillo et al., 2019b), and JKO-based schemes (Carrillo et al., 2022). For a comprehensive overview, see (Bailo et al., 2024b, §5). Recently, the research field was empowered by deep learning techniques. In particular, (Alvarez-Melis et al., 2022; Mokrov et al., 2021; Fan et al., 2022; Park et al., 2023; Lee et al., 2024) employ gradient-based optimization and neural-network parametrization to solve WGFs with given energy $\mathcal{J}$ as to special cases of (5).

### 4.2 LEARNING ENERGY FOR AGGREGATION-DIFFUSION EQUATION VIA JKO SCHEME

This line of research focuses on methods for *learning* population dynamics (§3.1) using the JKO scheme, as in `JKOnet` (Bunne et al., 2022b) and `JKOnet*` (Terpin et al., 2024). We discuss these methods in detail below, as they are the most closely related to our approach. Additional methods for learning population dynamics are reviewed in Appendix A.1.1.

**JKOnet** (Bunne et al., 2022b) formulates the task of population dynamics recovery as a bi-level optimization problem aimed at minimizing the discrepancy between observed distributions $\rho_k$ and model predictions $\hat{\rho}_k$:

$$\mathcal{L}_{\text{JKOnet}}(\theta, \varphi) = \sum_{k=0}^{K-1} d_{\mathbb{W}_2}^2(\hat{\rho}_k, \rho_k), \quad \text{s.t. } \hat{\rho}_0 = \rho_0, \quad \hat{\rho}_{k+1} = \nabla \psi_k^* \sharp \hat{\rho}_k,$$

$$\psi_k^* \stackrel{\text{def}}{=} \underset{\varphi: \psi_\varphi \in \text{CVX}}{\arg\min} \mathcal{J}_\theta(\nabla \psi_\varphi \sharp \hat{\rho}_k) + \frac{1}{2\tau} \int_\mathcal{X} \|x - \nabla \psi_\varphi(x)\|^2 \, d\hat{\rho}_k(x),$$

(16)

where CVX denotes the set of continuously differentiable convex functions from $\mathcal{X}$ to $\mathbb{R}$. The transport maps $\psi_\varphi$ are parametrized either using ICNNs (Bunne et al., 2022a) or through the 'Monge gap' approach (Uscidda & Cuturi, 2023). However, the energy functional is limited to the potential energy term $\mathcal{V}_\theta(\rho)$, excluding interaction and internal components. This framework has two key limitations: **(i)** solving the bi-level optimization problem in (16) is computationally challenging and requires specific techniques like unrolling optimizer's steps, and **(ii)** extending the method to more realistic settings with richer energy structures is questionable due to computational complexity.

**JKOnet\*** (Terpin et al., 2024) addresses the limitations of the original `JKOnet` by replacing the full JKO optimization problem (8) with its first-order optimality conditions. This reformulation allows

the use of more expressive energy functionals (12). The method minimizes the following objective, utilizing a *precomputed* optimal transport plans $\pi_k$ between $\rho_k$ and $\rho_{k+1}$:

$$
\mathcal{L}_{\texttt{JKOnet}^*}(\theta) = \sum_{k=0}^{K-1} \int_{\mathcal{X}\times\mathcal{X}} \left\| \nabla V_{\theta_1}(x_{k+1}) + \int_{\mathcal{X}} \nabla U_{\theta_2}(x_{k+1} - y_{k+1}) \, \mathrm{d}\rho_{k+1}(y_{k+1}) \right.
$$
$$
\left. + \theta_3 \frac{\nabla \rho_{k+1}(x_{k+1})}{\rho_{k+1}(x_{k+1})} + \frac{1}{\tau}(x_{k+1} - x_k) \right\|^2 \mathrm{d}\pi_k(x_k, x_{k+1}). \tag{17}
$$

The authors emphasize several advantages over `JKOnet`, including reduced computational costs and support for more general energy functionals. However, their method requires an additional optimization round to precompute optimal transport plans $\pi_k$, introducing extra sources of inaccuracy and rendering the approach non-end-to-end. Moreover, the authors of (Terpin et al., 2024) employ discrete OT solvers to compute $\pi_k$, which may fail to accurately represent OT maps between the underlying distributions, particularly in high-dimensional settings (Deb et al., 2021).

**iJKOnet (Ours)** integrates the strengths of both `JKOnet` and `JKOnet`*. From `JKOnet`, it inherits parameterized models for the energy functional $\mathcal{J}_\theta$ and the transport map $T_\varphi$, avoiding reliance on precomputed discrete optimal transport plans $\pi_k$, which can hinder flexible mappings between $\rho_k$ and $\rho_{k+1}$. From `JKOnet`*, it adopts a more expressive formulation of the energy functional (12) – capable of capturing complex structures. Unlike `JKOnet`*, our formulation does not rely on analytical expressions of the JKO step (8) optimality conditions; instead, it solves the step directly via the inner minimization in (11). This approach makes the method more readily extendable to general energy forms, such as porous medium energies (Alvarez-Melis et al., 2022), although exploring such extensions is beyond the scope of the current work.

## 5 EXPERIMENTS

Our implementation is primarily based on the publicly available code[1] from `JKOnet`* (Terpin et al., 2024), written in JAX (Bradbury et al., 2018), and available at the following repository[2]. Since our approach builds on the JKO framework, in §5.1, we first reproduce the experimental setup from `JKOnet`* for learning a known potential energy. Although we found the original codebase well-structured and accessible, we identified several inconsistencies in the data generation process, which we discuss in detail there. Finally, in §5.2, we compare our method against popular non-JKO baselines on single-cell RNA-seq datasets. Extended comparisons can be found in Appendix B.

**JKO-based Models.** As demonstrated in Appendix B.1.2 and B.1.3, learning the interaction and internal energy components directly from samples is challenging – likely because their estimation requires computing integrals over functions that must themselves be estimated. Consequently, for large-scale experiments, we restrict our model to the potential energy parametrization, imposing the inductive bias $\theta_2 = \theta_3 = 0$. We refer to this variant as $\texttt{iJKOnet}_V$. For moderate-dimensional tasks, we explore all combinations of parametrizations to illustrate the challenges of full energy recovery. For a fair comparison with (Terpin et al., 2024), we also implemented a time-varying potential energy parametrization, as described in Appendix B of the original paper, and denote this variant by $\texttt{iJKOnet}_{t,V}$. However, as discussed in Appendix A.2, this parametrization does not correspond to a single, consistent energy functional governing the dynamics. Instead, it actually solves a sequence of $K$ independent optimal transport problems between consecutive snapshots (Bunne et al., 2023). Moreover, because `JKOnet` (Bunne et al., 2022b) is computationally expensive, we limit our comparisons to experiments with moderate dimensionality, with the corresponding results reported in Appendix B.2. Technical details of model training are provided in Appendix C.

**Metrics.** We evaluate the next-step distribution $\hat{\rho}_{k+1}$ from $\rho_k$ and compare it to the ground-truth distribution $\rho_{k+1}$ using the following metrics: **(a)** Earth Mover's Distance (EMD) (or $\mathcal{W}_1$) (Rubner et al., 1998) and $\mathcal{W}_2$, **(b)** *Bures-Wasserstein Unexplained Variance Percentage* ($\mathrm{B}d_{\mathbb{W}_2}^2$-UVP) (Korotin et al., 2021c), **(c)** $\mathcal{L}_2$-*based Unexplained Variance Percentage* ($\mathcal{L}_2$-UVP) (Korotin et al., 2021a), which measures the discrepancy between the ground-truth functional $F^*$ and its reconstruction $\widehat{F}$, and **(d)** *Maximum Mean Discrepancy* (MMD) (Gretton et al., 2012). Formal definitions and implementation details for all metrics are provided in Appendix C.1.

---

[1] ⌂ https://github.com/antonioterpin/jkonet-star
[2] ⌂ https://github.com/MuXauJl11110/iJKOnet

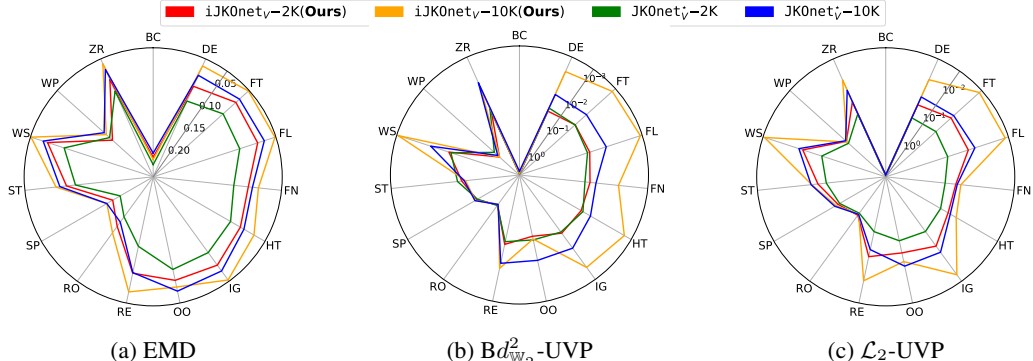

Figure 2: Numerical results from §5.1 for the **unpaired** setup. The reported absolute values show that, while increasing the number of samples generally improves performance across metrics, certain potentials remain challenging, highlighting the difficulty of this setup.

## 5.1 LEARNING POTENTIAL ENERGY

As discussed in §1, the population dynamics setting assumes that particle trajectories are not tracked across time – i.e., each time snapshot should consist of independently sampled particles without temporal correlation. This implies that data at each time step should be regenerated from scratch. However, we identified an inconsistency in the original codebase[1]: particle trajectories are preserved across time steps, resulting in temporally correlated samples. We refer to this as the **paired** setup, where each particle $x_k$ is directly linked to $x_{k+1}$ along a trajectory. In contrast, the intended, temporally uncorrelated setting is referred to as the **unpaired** setup. Table 3 in Appendix B.1.1 shows that switching between setups has a substantial impact on performance. A detailed discussion of this phenomenon is provided in Appendix A.3.

Following (Terpin et al., 2024, §4.1), we begin by examining how our method learns potentials in the 2D **paired** setup, which allows for a direct visual comparison with `JKOnet*`. These experiments are conducted on the synthetic dataset from (Terpin et al., 2024, Appendix B), with results presented in Figure 4 and further details provided in Appendix B.1.1. We then repeat the same procedure in the 2D **unpaired** setup, analyzing how performance scales with the number of samples (2K and 10K). The corresponding results are shown in Figure 2. They indicate that our method outperforms `JKOnet*` on nearly all potentials, and that for certain cases, increasing the number of samples does not improve performance, highlighting the difficulty of the unpaired setup.

## 5.2 LEARNING POPULATION DYNAMICS FOR SINGLE-CELL DATA

Following (Chen et al., 2023) and (Shen et al., 2025), in this section, we apply our method to modeling the dynamics of single-cell RNA data, with extended comparisons provided in Appendix B.2.

**Dataset.** We use the Embryoid Body (**EB**) dataset (Moon et al., 2019) and follow the preprocessing pipeline described in (Tong et al., 2020). The **EB** dataset comprises a collection of 5 timepoints and describes the differentiation of human embryonic stem cells over a period of 30 days.

**Non-JKO Models.** We evaluate our method against classic algorithms for trajectory inference such as `TrajectoryNet` (Tong et al., 2020), `MIOFLOW` (Huguet et al., 2022), `DMSB` (Chen et al., 2023), `NLSB` (Koshizuka & Sato, 2023) and the recently proposed `MMSB` (Shen et al., 2025). For these methods, the reported metrics are taken from the corresponding referenced papers.

**Leave-two-out in 5D.** Following (Shen et al., 2025), we perform experiments using a leave-two-out setup. Since the **EB** dataset contains five timesteps $\mathbf{t}_{(0-4)}$, we remove the first ($\mathbf{t_1}$) and third ($\mathbf{t_3}$) timesteps from training data (i.e., it consists of $\mathbf{t_{0,2,4}}$ timesteps), and then evaluate how well our method can reconstruct the data for $\mathbf{t_1}$ and $\mathbf{t_3}$ from the remaining $\mathbf{t_0}$ and $\mathbf{t_2}$ timesteps, respectively. The results are demonstrated in Table 1. One can see that our method with the "$t, V$" parameterization yields the best metrics, whereas the use of alternative parameterizations (with interaction and internal energies) results in poorer metrics. `Vanilla-SB` corresponds to the IPF (GP) method

(Vargas et al., 2021); further details are provided in Appendix B.6 of (Shen et al., 2025). Our method outperforms previous approaches in terms of the provided metric.

**Leave-one-out in 100D.** Following (Chen et al., 2023), we conduct experiments using a leave-one-out setup. One of the timesteps $t_1$, $t_2$, or $t_3$ is omitted, and we evaluate the method's ability to reconstruct the distribution of the left-out timestep. The results are shown in Table 2. Our method `iJKOnet`$_V$ achieves results on par with DMSB in the all-time (w/o LO) setting while using a simpler, simulation-free optimization routine that requires no trajectory caching. Consequently, it outperforms DMSB in execution time, as shown by comparing Table 12 in Appendix E of (Shen et al., 2025) (for 5D) with Table 7 (for 100D).

## 6 DISCUSSION

**Contributions.** We introduce the novel `iJKOnet` method, which provides a general framework for recovering any type of energy functional governing the evolution of a system; in this work, we focus on the free energy parametrization (12). Our work bridges the gap between inverse optimization theory and computational gradient flows. The proposed method outperforms previous JKO-based approaches in all comparisons and achieves comparable results to non-JKO methods on real-world tasks. We also establish theoretical guarantees for recovering dynamics driven by potential energy.

**Limitations.** Our method can not recover other types of internal energy functionals except for the entropy, nor can it handle time-dependent interaction energies, although theoretical results for such cases exist (Ferreira & Valencia-Guevara, 2018). Furthermore, it does not account for birth–death dynamics as in recent trajectory inference methods (Zhang et al., 2025b). The reliance on entropy estimation may also lead to performance degradation in higher-dimensional settings, and we find that learning interaction energies remains particularly challenging. As a result, jointly optimizing all energy parameters $(\theta_1, \theta_2, \theta_3)$ can cause instability and convergence to inaccurate potential estimates. Finally, from a theoretical standpoint, our analysis currently provides guarantees only for potential energy (Theorem (3.1)); extending it to other energy types is an avenue for future work.

Table 1: **5D** experiment. $d_{\mathbb{W}_2}$ distance ($\downarrow$) comparison across $t_1$ and $t_3$. Results for baselines (Shen et al., 2025).

| Method | $t_1$ | $t_3$ |
|---|---|---|
| Vanilla-SB | $1.49 \pm 0.063$ | $1.55 \pm 0.034$ |
| DMSB | $1.13 \pm 0.082$ | $1.45 \pm 0.16$ |
| TrajectoryNet | $2.03 \pm 0.04$ | $1.93 \pm 0.08$ |
| MMSB | $1.27 \pm 0.028$ | $1.57 \pm 0.048$ |
| **Static** | | |
| JKOnet$^*_V$ | $1.145 \pm 0.033$ | $2.529 \pm 0.014$ |
| JKOnet$^*_{V+U}$ | $1.099 \pm 0.119$ | $2.537 \pm 0.054$ |
| JKOnet$^*_{V+W}$ | $1.419 \pm 0.173$ | $2.510 \pm 0.094$ |
| JKOnet$^*_{W+U}$ | $1.887 \pm 0.017$ | $1.739 \pm 0.037$ |
| JKOnet$^*$ | $1.361 \pm 0.257$ | $2.557 \pm 0.042$ |
| **Static (Ours)** | | |
| iJKOnet$_V$ | $1.082 \pm 0.011$ | $1.147 \pm 0.001$ |
| iJKOnet$_{V+U}$ | $1.065 \pm 0.018$ | $1.150 \pm 0.004$ |
| iJKOnet$_{V+W}$ | $2.865 \pm 0.166$ | $1.376 \pm 0.015$ |
| iJKOnet$_{W+U}$ | $1.649 \pm 0.005$ | $0.868 \pm 0.005$ |
| iJKOnet | $3.577 \pm 0.166$ | $1.395 \pm 0.032$ |
| **Time-varying** | | |
| JKOnet$^*_{t,V}$ | $4.414 \pm 1.499$ | $2.771 \pm 0.197$ |
| iJKOnet$_{t,V}$ **(Ours)** | $\mathbf{0.983 \pm 0.037}$ | $\mathbf{0.849 \pm 0.021}$ |

Table 2: **100D** experiment. MMD distance ($\downarrow$). Comparison of methods across different leave-one-out splits. The results are averaged across 3 runs. Results for baseline methods: (Chen et al., 2023).

| Method | LO-$t_1$ | LO-$t_2$ | LO-$t_3$ | w/o LO |
|---|---|---|---|---|
| NLSB (Koshizuka & Sato, 2023) | 0.38 | 0.37 | 0.37 | 0.66 |
| MIOFLOW (Huguet et al., 2022) | 0.23 | 0.90 | 0.23 | 0.23 |
| DMSB (Chen et al., 2023) | $\mathbf{0.042 \pm 0.020}$ | $\mathbf{0.033 \pm 0.003}$ | $\mathbf{0.040 \pm 0.020}$ | $\mathbf{0.032 \pm 0.003}$ |
| JKOnet$^*_V$ (Terpin et al., 2024) | $0.220 \pm 0.025$ | $0.293 \pm 0.018$ | $0.235 \pm 0.006$ | $0.229 \pm 0.052$ |
| iJKOnet$_V$ **(Ours)** | $0.137 \pm 0.001$ | $0.123 \pm 0.001$ | $0.097 \pm 0.002$ | $0.085 \pm 0.024$ |
| JKOnet$^*_{t,V}$ (Terpin et al., 2024) | $0.575 \pm 0.119$ | $0.619 \pm 0.157$ | $0.456 \pm 0.056$ | $0.477 \pm 0.098$ |
| iJKOnet$_{t,V}$ **(Ours)** | $0.848 \pm 0.043$ | $0.370 \pm 0.233$ | $0.055 \pm 0.007$ | $0.124 \pm 0.243$ |

**Reproducibility.** We provide the experimental details in Appendix C and the code required to reproduce the conducted experiments is available in this repository (see `README.md` for details).

**LLM Usage.** Large Language Models (LLMs) were used exclusively to assist with sentence rephrasing and improving text clarity. All scientific content, results, and interpretations in this paper were produced solely by the authors.

## 7 ACKNOWLEDGMENTS

The work was supported by the grant for research centers in the field of AI provided by the Ministry of Economic Development of the Russian Federation in accordance with the agreement 000000C313925P4F0002 and the agreement №139-10-2025-033.

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

CONTENTS

## A  EXTENDED DISCUSSIONS

We first review related work in Appendix A.1 and then discuss the validity of time-varying potentials proposed by Terpin et al. (2024) in Appendix A.2 in contrast to the stationary setting considered in this paper. Finally, in Appendix A.3, we discuss potential reasons for the performance gap between the paired and unpaired setups introduced in Appendix 5.1.

### A.1  RELATED WORKS

In this section, we first provide an extended discussion of related work on learning population dynamics (also known as trajectory inference) in Appendix A.1.1, followed by an overview of applications of the JKO scheme (Jordan et al., 1998) in deep learning in Appendix A.1.2.

### A.1.1  LEARNING POPULATION DYNAMICS

In contrast to our approach, which models population dynamics as a Wasserstein gradient flow (WGF) (Ambrosio et al., 2008) of the free-energy functional (5) (Gómez-Castro, 2024), several other fruitful approaches have been proposed in the literature. We review these alternatives below.

**RNNs and Neural ODEs.** One of the first works on learning population dynamics is (Hashimoto et al., 2016), which used recurrent neural networks (RNNs) to learn the SDE (7) as a WGF of a potential energy functional. Later, (Chen et al., 2018a, CNF) introduced neural ODEs as the continuous-time limit of RNNs for learning ODE-based dynamics, and (Li et al., 2020) subsequently extended this approach to handle SDEs. Building on these ideas, (Erbe et al., 2023, `RNAForecaster`) applied neural ODEs to predict future transcriptomic states of single cells (Battich et al., 2020). More recently, (Li et al., 2025a) proposed `WeightFlow`, which models stochastic population dynamics by learning the continuous evolution of neural network weights that parameterize distributions at each time point, rather than modeling the dynamics directly in the state space. Their key idea is

to project the evolving distribution into the weight space of a backbone neural network trained to represent the probability distribution at each snapshot.

**Static OT.** Another line of work models population dynamics by directly learning $K$ *independent* transport maps from time-snapshot data, using the *static* optimal transport formulation (1) or (2), without attempting to recover the unifying energy functionals guiding the system evolution. For example, (Schiebinger et al., 2019, `Waddington-OT`) employs discrete transport maps, whereas (Bunne et al., 2023, `CellOT`) learns *neural* transport maps, as in (Fan et al., 2023, Eq.(6)), (Rout et al., 2022, Eq.(14)), and (Korotin et al., 2023b, Eq.(15)), see Appendix A.2 for more details. Such approaches are typically called Neural OT solvers (Korotin et al., 2021b; 2023a; Mokrov et al., 2024; Gushchin et al., 2024b; Tarasov et al., 2026; Asadulaev et al., 2024) and have recently obtained many different extensions to certain extended OT settings, in particular to the barycenter problem (Kolesov et al., 2024a; Gazdieva et al., 2025; Korotin et al., 2022; 2021c; Fan et al., 2021) which can also be relevant for modeling data evolution in time (Kolesov et al., 2024b, Appendix C.5). However, they are not that widespread in applications to population dynamics.

**Dynamic OT.** (Tong et al., 2020, `TrajectoryNet`) models population dynamics using the dynamic optimal transport formulation of (Benamou & Brenier, 2000), parameterized by a regularized CNF. The work (Huguet et al., 2022, `MIOFlow`) later extended this approach by incorporating a geodesic autoencoder to better capture manifold structure. More recently, (Wan et al., 2023) proposed computational techniques that make dynamic OT scalable to higher-dimensional settings.

**Flow Matching.** The works in this paragraph extend Flow Matching (FM) techniques (Lipman et al., 2023; Albergo & Vanden-Eijnden, 2023; Liu et al., 2023) in several directions. Conditional Flow Matching (Tong et al., 2024a, `CFM`) and Optimal Flow Matching (Kornilov et al., 2024, `OFM`) introduce simulation-free objectives for learning deterministic and optimal flows. Building on the celebrated work of (Otto, 2001), the Wasserstein space can be nominally (i.e., heuristically) viewed as a Riemannian manifold. Motivated by this observation, Wasserstein Flow Matching (Haviv et al., 2025, `WFM`) applies the idea of Riemannian Flow Matching (Chen & Lipman, 2024, `RFM`) to the Wasserstein space, i.e., performing FM over a distribution of distributions. Further developments, such as Meta Flow Matching (Atanackovic et al., 2025, `Meta FM`), embed the population of samples with a Graph Neural Network (GNN) (Liu et al., 2025) and use these embeddings as conditioning inputs for the learning vector field. Finally, Multi-Marginal Flow Matching (Rohbeck et al., 2025, `MMFM`) constructs a flow using smooth spline-based interpolation across time points and conditions, and regresses it with a neural network under the classifier-free guided Flow Matching framework (Zheng et al., 2023). For a broader overview of FM for learning population dynamics, see the recent surveys (Morehead et al., 2025; Li et al., 2025b).

**Action Matching.** (Neklyudov et al., 2023, `AM`) introduced Action Matching (AM), a method that optimizes an action-gap objective. In contrast to Flow Matching, AM learns a vector field expressed as the gradient of an action, $\nabla s_t^*$, which uniquely defines the velocity field that transports particles optimally in the sense of optimal transport. Building on this idea, (Berman et al., 2024) conditioned the action $s_{t,\mu}$ on physical parameters $\mu$, used (Berman & Peherstorfer, 2024, `CoLoRa`) for efficient action parametrization, and demonstrated its utility for surrogate modeling of classical numerical solvers, enabling fast prediction of system behavior across different physics parameter settings. Most recently, (Neklyudov et al., 2024, `WLF`) proposed Wasserstein Lagrangian Flows, a unifying framework that minimizes Lagrangian action functionals over the space of probability densities rather than the ground space, thereby encapsulating AM as a special case. Interestingly, action matching can be modified just like the flow matching (Kornilov et al., 2024, `OFM`) to obtain the solutions to optimal transport after training, see (Kornilov & Korotin, 2025, `OAM`).

**Schrödinger bridges.** Two seminal works, Vargas et al. (2021, `IPF (GP)`) and De Bortoli et al. (2021, `IPF (NN)`), proposed solving the Schrödinger bridge problem using the Iterative Proportional Fitting (IPF) algorithm (Gramer, 2000) in application to generative modeling, which alternates between forward and backward processes. The first approach employed Gaussian processes as parametrization, whereas the second relied on score-based neural networks, see (Gushchin et al., 2024c) for a quick survey and benchmark. Later, bridge matching methods were developed (Peluchetti, 2023b;a; Liu & Wu, 2023; Shi et al., 2023; Korotin et al., 2024; Gushchin et al., 2024d;a; Ksenofontov & Korotin, 2025; Kholkin et al., 2026; Tong et al., 2024b; Carrasco et al., 2026). Additional contributions include generalized setups for Schrödinger bridges (Koshizuka & Sato, 2023; Liu et al., 2024), multi-marginal generalizations of the problem (Chen et al., 2023, `DMSB`), (Shen

et al., 2025, `MMSB`), (Hong et al., 2025), and momentum-accelerated formulations (Theodoropoulos et al., 2025, `3MSBM`) which are particularly suitable for trajectory inference problems.

**Unbalanced OT.** Several works jointly model marginal transitions and growth dynamics by minimizing the action in the Wasserstein-Fisher-Rao (WFR) metric (Chizat et al., 2018a), i.e., by solving the dynamical unbalanced optimal transport problem (Chizat et al., 2018b). (Tong et al., 2023) proposed `BEMIOflow`, an extension of (Huguet et al., 2022, `MIOflow`), that incorporates a neural network to predict cell growth and death rates continuously in time, thereby augmenting optimal transport with population-size dynamics. Later, (Chen et al., 2022b) formulated an unbalanced Schrödinger Bridge problem, which was applied in practice by (Zhang et al., 2025b, `DeepRUOT`), following the developments in (Sha et al., 2024, `TIGON`). `DeepRUOT` was further simplified into a single-network formulation, `Var-RUOT` (Sun et al., 2025), by exploiting optimality conditions; extended to include interaction modeling in (Zhang et al., 2025d, `CytoBridge`); and adapted to use two independent networks to separately model distributional drift and mass growth (Wang et al., 2025, `VGFM`), (Zhang et al., 2025a). In a different direction, (Klein et al., 2024, `GENOT`) generalized OT to simultaneously handle stochasticity, entropy regularization, and unbalanced mass transport, enabling applications to cross-modal and heterogeneous data. For a broader survey of trajectory inference methods and their applications to biology data, see (Zhang et al., 2025e;c). For completeness, we note that there exists a line of works developing efficient solvers for unbalanced OT (Gazdieva et al., 2024b;a; Choi et al., 2023), but they do not consider applications to biological data.

The concurrent work (Andrade et al., 2025, `iJKO`) also connects inverse optimization with the JKO scheme (8) but in the unbalanced setting. However, their focus is on sample complexity, the method addresses only potential energy, and experiments are restricted to low-dimensional cases.

**Riemannian perspective.** (Scarvelis & Solomon, 2023) proposed learning a metric tensor $A(x)$ that minimizes the average 1-Wasserstein distance on the learned manifold between pairs of consecutive population snapshots. Furthermore, (Kapusniak et al., 2024, `MFM`) introduced Metric Flow Matching (MFM), which learns interpolants $x_{t,\eta} = (1-t)x_0 + tx_1 + t(1-t)\varphi_{t,\eta}(x_0, x_1)$, where $\eta$ are the parameters of a neural network $\varphi_{t,\eta}$ providing a nonlinear "correction" to straight-line interpolants (Albergo & Vanden-Eijnden, 2023). The resulting velocity field minimizes a data-dependent Riemannian metric, assigning lower transport cost to regions with higher data density.

**Splines in Wasserstein Space.** In the seminal work of (Schiebinger et al., 2019), a piecewise linear OT interpolation method was proposed to infer cell trajectories. Subsequent works introduced higher-order piecewise polynomials (e.g., cubic splines) in Wasserstein space: (Chen et al., 2018b; Benamou et al., 2019) formulated a global cubic spline minimization problem, which was later extended by (Chewi et al., 2021) to use Euclidean interpolation algorithm in $\mathbb{R}^D$ after a finding optimal Monge map (2) between samples from consecutive measures, with further refinements by (Clancy & Suarez, 2022; Justiniano et al., 2024). More recently, (Banerjee et al., 2025, `WLR`) proposed the Wasserstein Lane–Riesenfeld method, leveraging the classical subdivision algorithm of (Lane & Riesenfeld, 1980). In parallel, (Dyn & Sharon, 2025) developed subdivision schemes for general metric spaces, including the Wasserstein space. (Baccou & Liandrat, 2024; Kawano et al., 2025) combines subdivision schemes with OT.

**Dynamics from stationary distributions.** A recent line of research aims to recover meaningful dynamical structure from static data alone. The Equilibrium Flow framework of (Zhang & Levin, 2025) addresses this problem by learning a vector field whose flow preserves the observed stationary distribution $p$. Their method enforces the invariance condition $\nabla \cdot [p(x,t)v(x,t)] = 0$ using score-based estimates of $\nabla \log p$, enabling the recovery of plausible dynamics even without temporal information or trajectories. Despite this minimal supervision, the learned flows can be used for inverse design, where one constructs dynamics that yield a desired target distribution.

### A.1.2   JKO SCHEME MEETS DEEP LEARNING

**Generative Modeling.** (Vidal et al., 2023, `JKO-Flow`) and (Xu et al., 2023, `JKO-iFlow`) reinterpret the JKO scheme (8) through the framework of CNFs (Chen et al., 2018b), applying it to 2D synthetic data and image generation tasks, respectively. Subsequently, (Cheng et al., 2024) provided theoretical convergence guarantees for this approach. (Choi et al., 2024, `S-JKO`) further accelerated WGF-based generative modeling by leveraging a semi-dual unbalanced OT formulation. They constructed the WGF of an $f$-divergence $D_f$ and used an (Song et al., 2021, `NCSN++`) backbone –

similar to (Zhu et al., 2024, NSGF), who designed a generative model as the WGF with respect to the Sinkhorn divergence (Peyré et al., 2019).

**Tensor Train (TT) (Oseledets, 2011).** (Aksenov & Eigel, 2025) introduces a method for approximating probability distributions in Bayesian inversion by minimizing a KL-based functional via an entropically regularized JKO scheme. The approach discretizes the resulting coupled heat equations on a high-dimensional grid and solves them using accelerated fixed-point methods combined with low-rank TT representations. Related work includes (Chertkov & Oseledets, 2021), which applies TT approximations to the Fokker-Planck equation with drift and diffusion, and (Han et al., 2025), which employs entropy-regularized proximal steps with TT-cross for particle-based evolution.

**Variational Inference.** (Lambert et al., 2022) and (Diao et al., 2023) study variational inference through the lens of Bures–Wasserstein gradient flows. (Cheng et al., 2023, GWG) introduce a generalized minimizing movement scheme on the space $\mathcal{P}_{c_h}(\mathbb{R}^D)$, where the transport cost is defined as $c_h(x, y) = g\left(\frac{x-y}{h}\right) h$ with $g$ a continuously differentiable Young function, and apply this framework to particle-based variational inference.

**Datasets/Weights Learning.** (Bonet et al., 2025) propose a explicit scheme that is computationally more efficient in practice than the implicit JKO scheme on the space $\mathcal{P}_2(\mathcal{P}_2(\mathbb{R}^D))$, and apply this approach to modeling flow datasets viewed as random measures. (Saragih et al., 2025) train a meta-model, based on JKO as well as other flow- and diffusion-based approaches, that generates dataset-conditioned classifiers by producing the neural network weights in a single forward pass.

## A.2 Time-varying potentials discussion

In this section, we examine the validity of using time-varying potentials, as proposed in (Terpin et al., 2024, §4.4). We demonstrate that under this formulation, the original problem of reconstructing energy functionals (Bunne et al., 2022b) via JKO Scheme reduces to learning $K$ independent neural optimal transport maps (Fan et al., 2023, Eq.(6)), (Rout et al., 2022, Eq.(14)) between the data snapshots $\rho_k$ and $\rho_{k+1}$. To illustrate this, consider the loss in (11) and assume that, instead of a single functional $\mathcal{J}$, we now assign a separate only potential energy functional $\mathcal{J}_{\text{PE}}^k = \int_{\mathcal{X}} V^k(x) \, \mathrm{d}\rho_k(x)$ for each time step $k$:

$$\max_{\mathcal{J}_{\text{PE}}^k} \min_{T^k} \mathcal{L}(\mathcal{J}^k, T^k) \stackrel{\text{def}}{=} \max_{\mathcal{J}_{\text{PE}}^k} \min_{T^k} \sum_{k=0}^{K-1} \left[ \mathcal{J}_{\text{PE}}^k(T^k \sharp \rho_k) - \mathcal{J}_{\text{PE}}^k(\rho_{k+1}) + \frac{1}{2\tau} \int_{\mathcal{X}} \|x - T^k(x)\|_2^2 \rho_k(x) \, \mathrm{d}x \right].$$

It then becomes clear that the loss optimization decomposes into $K$ independent terms $\mathcal{L}^k(\mathcal{J}_{\text{PE}}^k, T^k)$:

$$\mathcal{L}^k(\mathcal{J}_{\text{PE}}^k, T^k) \stackrel{\text{def}}{=} \mathcal{J}_{\text{PE}}^k(T^k \sharp \rho_k) - \mathcal{J}_{\text{PE}}^k(\rho_{k+1}) + \frac{1}{2\tau} \int_{\mathcal{X}} \|x - T^k(x)\|_2^2 \rho_k(x) \, \mathrm{d}x \to \max_{\mathcal{J}_{\text{PE}}^k} \min_{T^k}. \quad (18)$$

By replacing $V^k$ with the dual potential $f_\eta$ and $T^k$ with $T_\theta$, we recover Equation (6) from (Fan et al., 2023), which aligns with the objective proposed in (Bunne et al., 2023, Eq. (9)) for modeling single-cell dynamics. Thus, (Terpin et al., 2024) introduces a new strategy for minimizing this loss rather than directly addressing the recovery of ground-truth energy functionals (Bunne et al., 2022b).

Fortunately, the theoretical framework developed by Ferreira & Valencia-Guevara (2018) and later extended by Plazotta & Zinsl (2016) establishes the **validity** of this approach even for time-varying free energy functionals (5), not only for the potential energy case, in the limit as the time discretization tends to zero. However, this connection is not acknowledged in (Terpin et al., 2024).

## A.3 Paired vs. unpaired setups

In this section, we discuss the substantial performance gap observed between the paired and unpaired setups, as illustrated in Table 3. We suggest that this discrepancy stems from the intrinsic connection between the considered solvers, iJKOnet and JKOnet*, and the underlying OT problem (2).

Both solvers are designed to recover the optimal transport mappings $T^{k,*}$ between consecutive distributions $\rho_k$ and $\rho_{k+1}$, satisfying the pushforward relation $T_{\mathcal{J}}^{k,*} \sharp \rho_k = \rho_{k+1}$. In JKOnet*, these mappings are obtained explicitly by precomputing the OT maps between $\rho_k$ and $\rho_{k+1}$, whereas in our method, they are learned implicitly via neural networks $T_\varphi^k$.

In practice, the continuous distributions $\rho_k$ are replaced by their empirical counterparts, constructed from training samples $\{x_k^1, x_k^2, \ldots, x_k^N\} = X_k \sim \rho_k$. This is precisely where the distinction between the paired and unpaired setups becomes critical:

- **Paired.** When samples are paired, they follow the ground-truth transport mappings, i.e.,

$$T^{0,*}(x_0^i) = x_1^i, \quad T^{1,*}(x_1^i) = x_2^i, \quad \ldots, \quad T^{k,*}(x_k^i) = x_{k+1}^i, \quad \ldots$$

  In this case, estimating $T^{k,*}$ becomes substantially simpler and can be viewed as a supervised regression problem over the given sample pairs.

- **Unpaired.** In contrast, when the samples $X_k$, $k \in \{0, \ldots, K\}$, are mutually independent, the quality of the estimated OT maps degenerate significantly. This degradation stems from the high sample complexity of Wasserstein-2 distances and maps (known to be relatively poor, see Hütter & Rigollet (2021)), as well as from the increased variance of the estimated transport mappings.

A deeper theoretical analysis of this gap between paired and unpaired setups remains an interesting direction for future work. To quantify how performance changes when transitioning from unpaired to paired data, we conducted the following *ablation study*.

**Experimental Setup.** Following Appendix B.1.1, we used the same set of potentials and focused solely on the potential parametrization. In the $2D$ setting with $K = 5$, we trained for $5K$ iterations using $N = 2K$ training samples per time step and a 40% test split. The fraction of full trajectories in the training set was gradually increased from 0.0 (unpaired) to 1.0 (paired) in steps of 0.25.

**Results.** Figure 3 shows the results. As the fraction of paired samples decreases, the performance of `JKOnet*` degrades more rapidly than that of our method. For the Bohachevsky potential, both methods show reduced accuracy in both paired and unpaired scenarios.

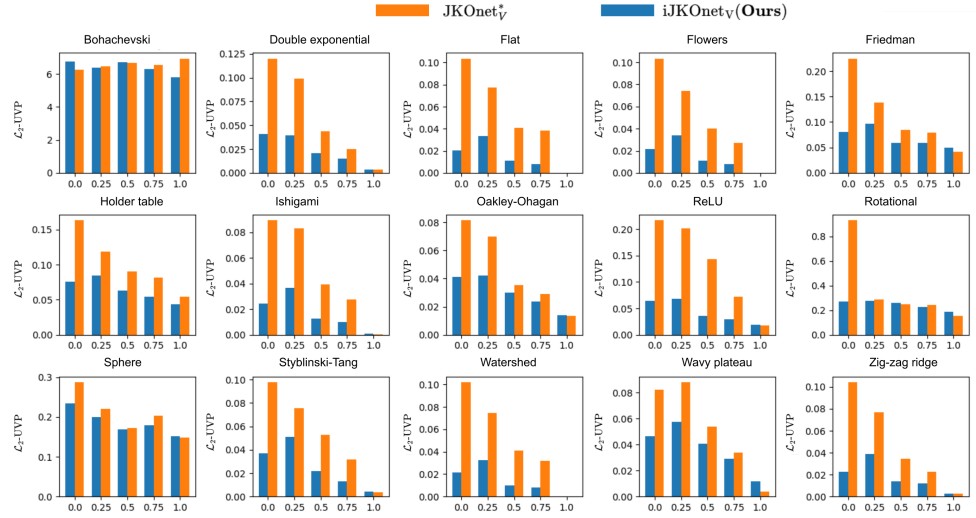

Figure 3: Ablation study comparing *paired* (1.0) and *unpaired* (0.0) setups. As the proportion of paired samples (i.e. full trajectories in the dataset) decreases, the performance of `JKOnet*` degenerate more rapidly than that of our method.

# B    EXTENDED COMPARISONS

In this section, we first present extended synthetic experiments in Appendix B.1, followed by results on real-world single-cell data in Appendix B.2.

## B.1 SYNTHETIC COMPARISONS

In this section, we further investigate our method's ability to learn potential energy (Appendix B.1.1), interaction energy (Appendix B.1.2), and internal energy (Appendix B.1.3).

Since our approach builds on the JKO framework for modeling population dynamics, we adopt the experimental setup from `JKOnet*` and compare our method against both `JKOnet` Bunne et al. (2022b) and `JKOnet*` Terpin et al. (2024) on corresponding synthetic evaluation tasks.

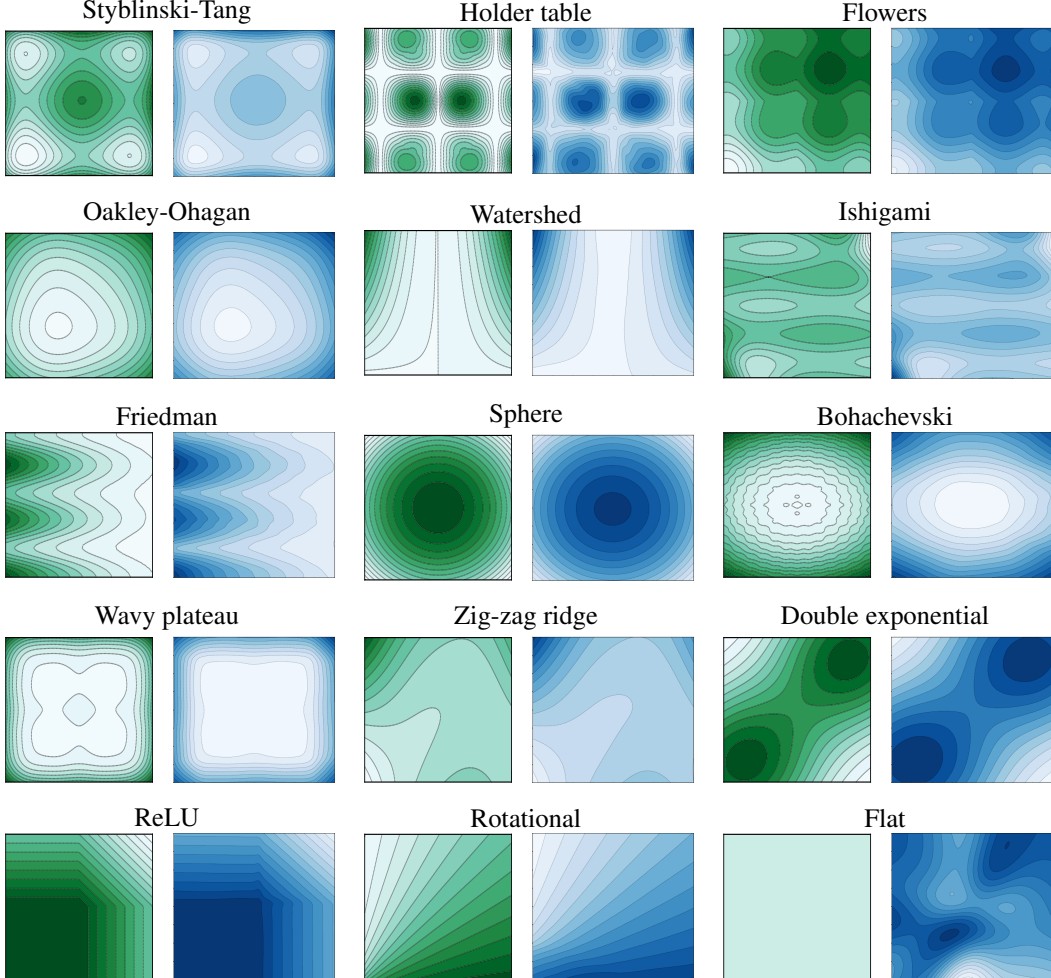

Figure 4: Level curves of the true (green) and estimated (blue) potentials reconstructed by **our** proposed `iJKOnet`$_V$ method for the **paired** setup, following (Terpin et al., 2024, Appendix F). These results can be directly compared with those in (Terpin et al., 2024, Figure 6). Note that for the *flat* potential, the value range is near zero, as expected for the ground-truth potential.

### B.1.1 LEARNING POTENTIAL ENERGY

**Paired Setup.** Following Terpin et al. (2024, §4.1), we evaluate our method in the paired setup (see details in §5.1), which enables a direct visual comparison with `JKOnet*` (Terpin et al., 2024). The experiments are conducted on the synthetic dataset from Terpin et al. (2024, Appendix B), using a step size of $\tau = 0.01$, $K = 5$ time steps, and $N = 2000$ samples per step with 40% of test samples. Figure 4 presents the ground-truth potentials $V(x)$ (green), defined in Terpin et al. (2024, Appendix F, Eqs. (31)–(45)), alongside the reconstructed potentials $V_\theta(x)$ (blue) learned by our method. As shown, our approach achieves performance comparable to `JKOnet*` in the paired setup.

**Unpaired Setup.** We repeat the experiments on a corrected version of the synthetic dataset, where samples are not temporally correlated. We select the six most challenging potentials in terms of convergence. Table 3 reports the ratio of the final metrics between the unpaired and paired setups. As shown, switching to the unpaired setup significantly affects the method's performance.

Table 3: Comparison of paired and unpaired setups. Each value indicates the ratio of the final metric in the unpaired setup to that in the paired setup. Most ratios significantly exceed 1, highlighting the increased difficulty of the unpaired setting.

| | | `iJKOnet`$_V$ **(Ours)** | | | `JKOnet`$^*_V$ | | |
|---|---|---|---|---|---|---|---|
| Alias | Potential Name | B$d^2_{\mathbb{W}_2}$-UVP | EMD | $\mathcal{L}_2$-UVP | B$d^2_{\mathbb{W}_2}$-UVP | EMD | $\mathcal{L}_2$-UVP |
| FL | Flowers | 8066 | 90 | 297 | 135325 | 305 | 5809 |
| FN | Friedman | 556 | 5 | 2 | 263 | 6 | 3 |
| IG | Ishigami | 2259 | 38 | 36 | 4276 | 64 | 210 |
| WS | Watershed | 18135 | 220 | 1509 | 512720 | 1014 | 102592 |
| WP | Wavy plateau | 2 | 2 | 7 | 1 | 2 | 29 |
| ZR | Zigzag ridge | 158 | 23 | 13 | 155 | 25 | 31 |

### B.1.2 LEARNING INTERACTION ENERGY

In this section, we evaluate how accurately our method and `JKOnet`$^*$ (Terpin et al., 2024) can approximate the interaction energy in the unpaired setup. We exclude `JKOnet` from comparison, as its design does not support learning interaction energies.

We identified an inconsistency in the theoretical formulation presented in the `JKOnet`$^*$ paper (Terpin et al., 2024). Specifically, as discussed in §2 (see also (Santambrogio, 2015, §7.2)), the ground-truth interaction functional must be symmetric; that is, in (5), the interaction kernel should satisfy $W(z) = W(-z)$. Incorporating this correction, we conducted an experiment to assess the ability of both methods to recover the *interaction* energy in a 2D unpaired setting.

**Experimental Setup.** The ground-truth interaction kernels are defined as $W(z) = \frac{1}{2}\big(W_b(z) + W_b(-z)\big)$, where $W_b$ denotes the base functionals listed in Table 3. Both methods were restricted to use only the interaction energy component $\mathcal{W}_{\theta_2}$ in the energy parameterization (12). The corresponding variants are denoted as `iJKOnet`$_W$ and `JKOnet`$^*_W$. Similar to §B.1.1, we use a step size of $\tau = 0.01$, $K = 5$ time steps, and two sample size settings: a small-scale setup with 12K total samples ($N = 2K$ per step, with 40% reserved for testing) and a large-scale setup with 60K total samples ($N = 10K$ per step, with 40% reserved for testing).

**Results.** Qualitative results are shown in Figure 5 for the paired setup and in Figure 6 for the unpaired setup, with quantitative $\mathcal{L}_2$-UVP metrics reported in Table 4 (see §C.1 for details). In the paired setup, most potentials are accurately restored, although Waby Plateau, Friedmann, and Flowers are not. In the unpaired setup, neither method successfully recovers the interaction energy, likely due to biases introduced by the batched estimation of the interaction term $\mathcal{W}_{\theta_2}$ in (12), which involves squaring the input. This approximation seems to cause both models to converge to batch-specific minima.

Table 4: Quantitative $\mathcal{L}_2$-UVP results for the **unpaired** setup in 2D interaction energy learning. Increasing the number of samples improves performance, though the effect is relatively limited.

| | Sample size: 2K | | Sample size: 10K | |
|---|---|---|---|---|
| **Potential** | `iJKOnet`$_W$ **(Ours)** | `JKOnet`$^*_W$ | `iJKOnet`$_W$ **(Ours)** | `JKOnet`$^*_W$ |
| Flowers | 0.0032 | 0.0101 | 0.0009 | 0.0069 |
| Friedman | 0.0087 | 0.0010 | 0.0024 | 0.0093 |
| Ishigami | 0.0046 | 0.0087 | 0.0008 | 0.0066 |
| Watershed | 0.0043 | 0.0086 | 0.0009 | 0.0081 |
| Zigzag Ridge | 0.0060 | 0.0104 | 0.0017 | 0.0034 |

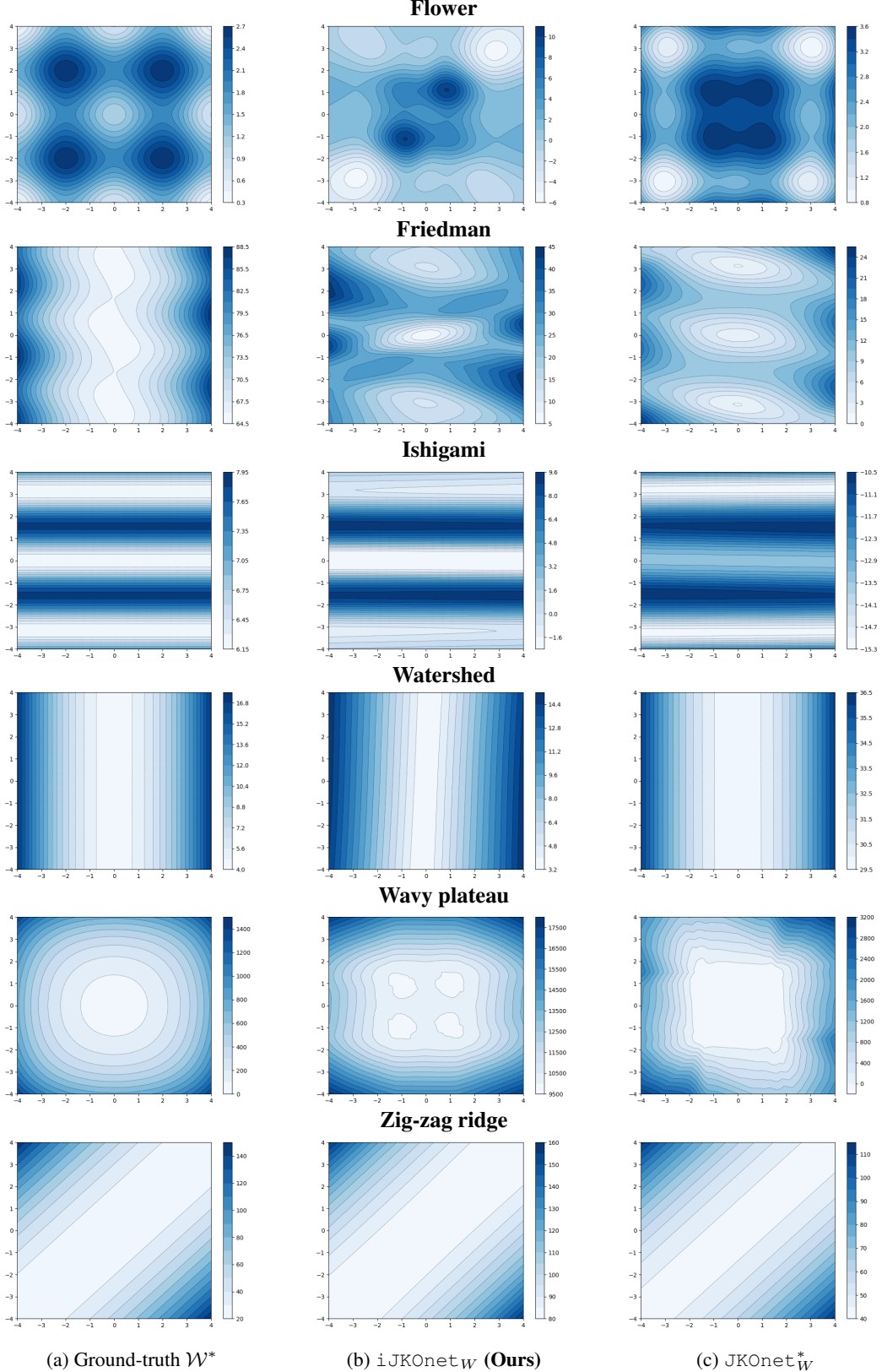

(a) Ground-truth $\mathcal{W}^*$        (b) iJKOnet$_W$ **(Ours)**        (c) JKOnet$_W^*$

Figure 5: Qualitative results from §B.1.2 in the **paired** setup, using 10K samples per time step (with 40% reserved for testing). Compared to Figure 6, which shows the unpaired setup, the model performs noticeably better: nearly all potentials are accurately reconstructed, highlighting the relative simplicity of the paired scenario.

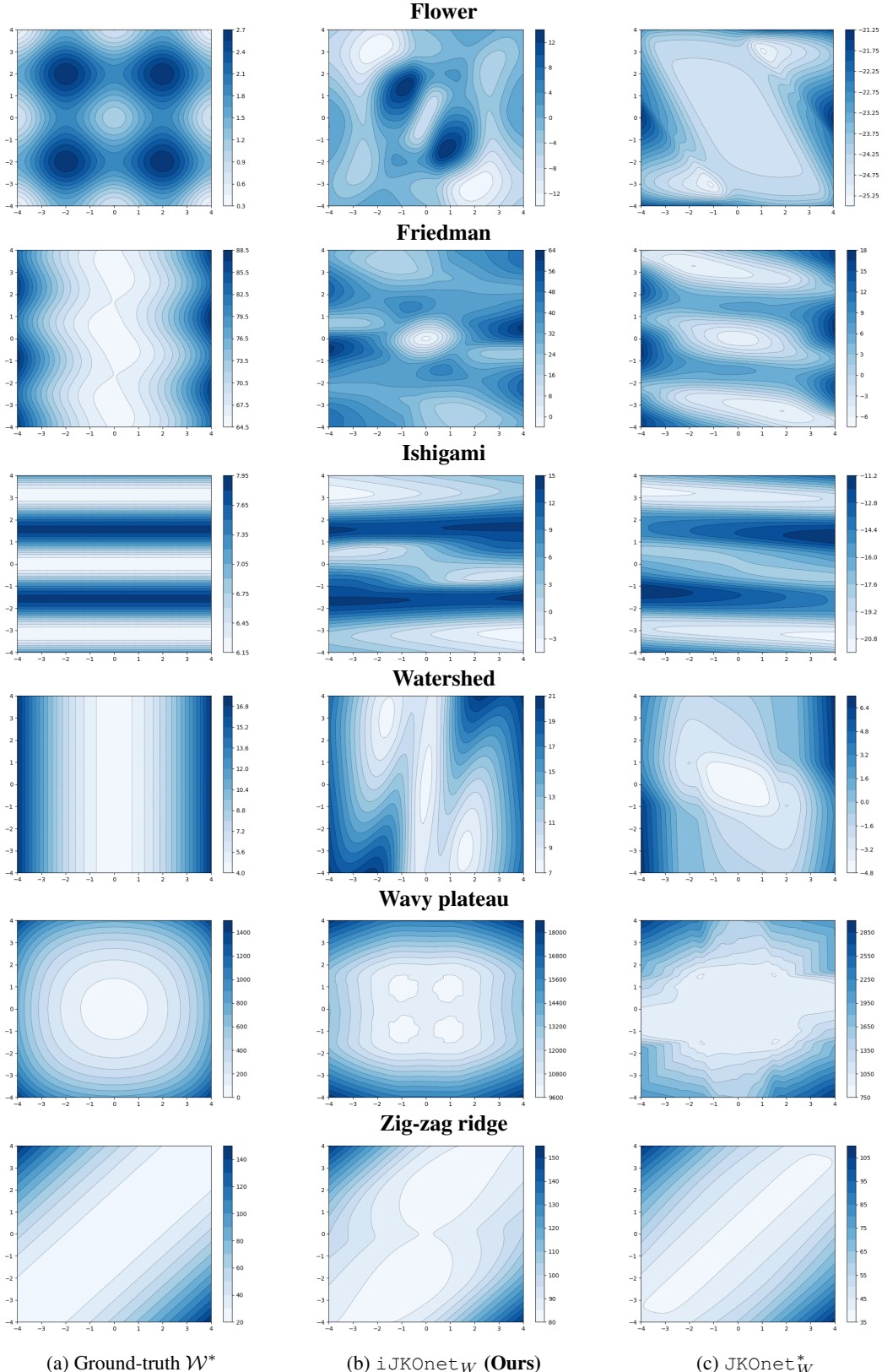

(a) Ground-truth $\mathcal{W}^*$      (b) iJKOnet$_W$ **(Ours)**      (c) JKOnet$_W^*$

Figure 6: Qualitative results from §B.1.2 in the **unpaired** setup with 10K samples in total (40% reserved for testing). Neither model is able to accurately reconstruct the ground-truth interaction energy $\mathcal{W}^*$, likely due to biases introduced in the estimation of the integral in (12).

### B.1.3 LEARNING INTERNAL ENERGY

In this section, we assess how accurately our method and $\text{JKOnet}^*$ (Terpin et al., 2024) can estimate the diffusion coefficient $\theta_3$ in (12) corresponding to the internal energy term under the **unpaired** setup. We exclude $\text{JKOnet}$ from comparison, as its design does not support learning internal energy.

**Experimental Setup.** Following the protocol of Terpin et al. (2024, §4.3), we estimate the diffusion coefficient $\theta_3$ in 2D and 20D settings using both $\text{iJKOnet}$ and $\text{JKOnet}^*$, i.e., the full parameterization in (12). The same functional from Appendix C.7 is used for both potential and interaction energies. The ground-truth diffusion levels are set to $\beta^* \in \{0.0, 0.1, 0.2\}$.

**Results.** Figure 7 shows that $\text{iJKOnet}$ fails to recover the ground-truth $\beta^*$ values, while $\text{JKOnet}^*$ provides a closer approximation. However, all predicted $\theta_3$ values tend to converge toward 0, deviating from the true levels $\beta^* \in \{0.0, 0.1, 0.2\}$. This indicates that accurately learning the internal energy remains challenging for both models. We assume that such behavior stems from difficulties of entropy estimation developed methods.

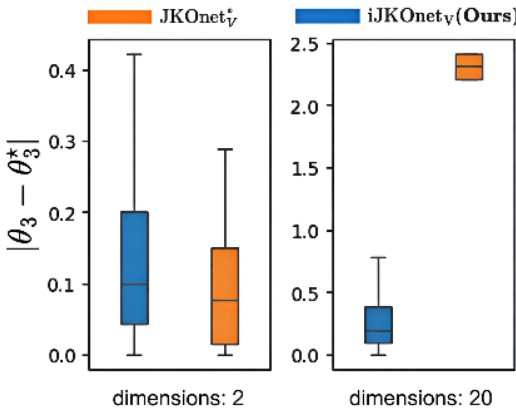

Figure 7: Estimation error of the diffusion coefficient $\theta_3$ relative to the ground-truth values $\beta^*$. Blue bars correspond to $\text{iJKOnet}$ and orange bars to $\text{JKOnet}^*$. The Y-axis indicates the absolute deviation between the estimated $\theta_3$ and true $\beta^*$ values.

### B.2 LEARNING SINGLE-CELL DYNAMICS

In this section, we present extended comparisons on real-world single-cell datasets, starting with 5D results and then considering 50D and 100D.

**Datasets.** Following §5.2, we consider the Embryoid Body (**EB**) dataset (Moon et al., 2019) and additionally the Multiome (**Multi**) dataset (Burkhardt et al., 2022). For both datasets, we apply the preprocessing pipeline of (Tong et al., 2020). The Multi dataset contains single-cell measurements across four time points (days 2, 3, 4, and 7), which we use for high-dimensional experiments.

**5D.** Following (Neklyudov et al., 2024, Table 2) and (Terpin et al., 2024, §4.4), we conducted experiments on the **EB** dataset in 5D. At each time step, we train for 1000 epochs on 60% of the data and compute the $d_{\mathbb{W}_1}$ distance between the observed distribution $\rho_k$ (remaining 40% of the data) and the one-step-ahead prediction $\hat{\rho}_k$. The results are shown in Table 5. Our method outperforms several previous approaches when using a time-varying potential parametrization.

**50D and 100D data.** Following Neklyudov et al. (2024), we conducted experiments on the **Multi** dataset in 50D and 100D using a leave-one-out setup, averaged over three runs. Models were trained on marginals from timepoint partitions [2,4,7] and [2,3,7], and evaluated on the corresponding left-out marginals at timepoints [3] and [4]. Training was performed for 5000 epochs.

The results are presented in Table 6. Our method achieves comparable performance for the left-out [3] marginal. However, performance on the [4] marginal is poor, leading to a lower overall average. This is likely due to the longer time intervals between the learned marginals, which violate our

Table 5: The results for the **EB** dataset in 5D are obtained by training on all time steps, evaluated for next-time-step prediction using the $d_{\mathbb{W}_1}$ metric. Results for non-JKO methods are taken from (Neklyudov et al., 2024, Table 2):

| Model | $t_0$ | $t_1$ | $t_2$ | $t_3$ | **Mean** |
|---|---|---|---|---|---|
| Neural SDE (Li et al., 2020) | 0.69 | 0.91 | 0.85 | 0.81 | 0.82 |
| TrajectoryNet (Tong et al., 2020) | 0.73 | 1.06 | 0.90 | 1.01 | 0.93 |
| SB-FBSDE (Chen et al., 2022a) | 0.56 | 0.80 | 1.00 | 1.00 | 0.84 |
| NLSB (Koshizuka & Sato, 2023) | 0.68 | 0.84 | 0.81 | 0.79 | 0.78 |
| OT-CFM (Tong et al., 2024a) | 0.78 | 0.76 | 0.77 | 0.75 | 0.77 |
| WLF-OT (Neklyudov et al., 2024) | 0.65 | 0.78 | 0.76 | 0.75 | 0.74 |
| WLF-SB (Neklyudov et al., 2024) | 0.63 | 0.79 | 0.77 | 0.74 | 0.73 |
| JKOnet (Bunne et al., 2022b) | 1.53 | 1.27 | 1.13 | 1.41 | 1.34 |
| JKOnet$^*_V$ Terpin et al. (2024) | 0.99 | 1.11 | 1.06 | 1.30 | 1.12 |
| iJKOnet$_V$ **(Ours)** | 0.92 | 1.11 | 0.95 | 1.21 | 1.05 |
| JKOnet$^*_{t,V}$ (Terpin et al., 2024) | 0.69 | 0.77 | 0.69 | 0.78 | 0.73 |
| iJKOnet$_{t,V}$ **(Ours)** | 0.51 | 0.58 | 0.57 | 0.64 | **0.58** |

assumption that consecutive marginals are obtained from JKO steps (see §3.1), thereby reducing the accuracy of predicting the [4] marginal.

Table 6: The results for **Multi** dataset for direct comparison with from (Neklyudov et al., 2024, Table 1), averaged for 3 runs for leave-one-out setup, $d_{\mathbb{W}_2}$ metric.

| Dimension | 50 | | | 100 | | |
|---|---|---|---|---|---|---|
| Metric | BW-UVP | EMD | MMD | BW-UVP | EMD | MMD |
| JKOnet$^*_V$ | $128.729 \pm 78.307$ | $68.406 \pm 6.055$ | 0.0003 | $92.512 \pm 38.075$ | $72.639 \pm 3.161$ | 0.0002 |
| iJKOnet$_V$ **(Ours)** | $38.318 \pm 18.325$ | $50.560 \pm 7.187$ | 0.0003 | $38.991 \pm 16.918$ | $59.216 \pm 6.869$ | 0.0002 |
| JKOnet$^*_{t,V}$ | $89.264 \pm 38.372$ | $68.600 \pm 9.846$ | 0.0003 | $93.429 \pm 34.332$ | $78.674 \pm 8.236$ | 0.0002 |
| JKOnet$_{t,V}$ **(Ours)** | $36.156 \pm 16.515$ | $50.026 \pm 7.727$ | 0.0003 | $38.387 \pm 18.525$ | $59.318 \pm 7.365$ | 0.0002 |

## C  EXPERIMENTAL DETAILS

### C.1  METRIC COMPUTATION DETAILS

In this section, we provide details on the evaluation metrics used in our experiments.

**Estimation protocol.** All metrics are computed using mini-batch Monte Carlo estimation. For synthetic experiments, we use 250 samples per time step. For real-world datasets, we use all available samples (approximately 500 per time step).

$\mathcal{L}_2$**-UVP.** When the ground-truth functional $F^*$ (e.g., $V$ or $W$) is available, we assess the discrepancy between it and its reconstruction $\widehat{F}$ using the *backward $\mathcal{L}_2$-based Unexplained Variance Percentage* ($\mathcal{L}_2$-UVP) metric introduced by Korotin et al. (2021a), defined as follows:

$$\mathcal{L}_2\text{-UVP}(F^*, \widehat{F}) = 100 \cdot \frac{\tau^2 \|\nabla \widehat{F} - \nabla F^*\|^2_{\rho_{k+1}}}{\text{Var}(\rho_k)}\%, \tag{19}$$

where the norm $\| \cdot \|_{\rho_{k+1}}$ is computed with respect to the ground-truth distribution $\rho_{k+1}$, and $\tau$ denotes the time step size. Values close to 0% indicate that $\nabla \widehat{F}$ closely approximates $\nabla F^*$.

$\mathbf{B}d^2_{\mathbb{W}_2}$**-UVP.** The *Bures-Wasserstein UVP* introduced by (Korotin et al., 2021c) is defined as

$$\mathrm{B}d^2_{\mathbb{W}_2}\text{-UVP}(\rho_k, \hat{\rho}_k) \overset{\text{def}}{=} 100 \cdot \frac{\mathrm{B}d^2_{\mathbb{W}_2}(\rho_k, \hat{\rho}_k)}{\frac{1}{2}\text{Var}(\rho_k)}\%, \tag{20}$$

where the *Bures-Wasserstein distance* is given by

$$\mathrm{B}d_{\mathbb{W}_2}^2(\mathbb{P}, \mathbb{Q}) \stackrel{\text{def}}{=} d_{\mathbb{W}_2}^2\left(\mathcal{N}(\mu_{\mathbb{P}}, \Sigma_{\mathbb{P}}), \mathcal{N}(\mu_{\mathbb{Q}}, \Sigma_{\mathbb{Q}})\right), \tag{21}$$

with $\mu_{\mathbb{P}}$ and $\Sigma_{\mathbb{P}}$ denoting the mean and covariance of distribution $\mathbb{P}$, respectively.

**EMD or $d_{\mathbb{W}_1}$.** The *Earth Mover's Distance* (EMD) (Rubner et al., 1998) is defined as

$$\mathrm{EMD}(\rho_k, \hat{\rho}_k) \equiv d_{\mathbb{W}_1}(\rho_k, \hat{\rho}_k) \stackrel{\text{def}}{=} \min_{\pi \in \Pi(\rho_k, \hat{\rho}_k)} \int_{\mathcal{X} \times \mathcal{X}} \|x - y\| \, \mathrm{d}\pi(x, y), \tag{22}$$

where $\rho_k$ and $\hat{\rho}_k$ denote the ground truth and predicted distributions at time step $k$.

In practice, we approximate the continuous formulation (22) using its discrete empirical counterpart (Peyré et al., 2019) and compute it with the POT library (Flamary et al., 2024). It is well known that empirical Wasserstein distances suffer from poor sample complexity in high dimensions (Fournier & Guillin, 2015; Panaretos & Zemel, 2019), meaning that a large number of samples is required for accurate estimation. In our experiments (up to 100 dimensions), the available sample sizes were sufficient to obtain stable estimates.

**MMD.** We compute the *Maximum Mean Discrepancy* (MMD) (Gretton et al., 2012) using its kernel formulation. Let $K : \mathcal{X} \times \mathcal{X} \to \mathbb{R}$ be a positive definite kernel with associated reproducing kernel Hilbert space (RKHS) $\mathcal{H}$ and feature map $\phi$. The squared MMD between distributions $\rho_k$ and $\hat{\rho}_k$ is

$$\mathrm{MMD}_K^2(\rho_k, \hat{\rho}_k) = \mathbb{E}_{x,x' \sim \rho_k} K(x, x') - 2\mathbb{E}_{x \sim \rho_k, y \sim \hat{\rho}_k} K(x, y) + \mathbb{E}_{y,y' \sim \hat{\rho}_k} K(y, y'). \tag{23}$$

Importantly, MMD can be computed directly via kernel evaluations without explicitly constructing feature maps. Given samples $\{x_i\}_{i=1}^N \sim \rho_k$ and $\{y_j\}_{j=1}^M \sim \hat{\rho}_k$, we use the unbiased estimator

$$\widehat{\mathrm{MMD}}_K^2 = \frac{1}{N(N-1)} \sum_{i \neq i'} K(x_i, x_{i'}) - \frac{2}{NM} \sum_{i=1}^N \sum_{j=1}^M K(x_i, y_j) + \frac{1}{M(M-1)} \sum_{j \neq j'} K(y_j, y_{j'}). \tag{24}$$

This estimator has quadratic complexity in the number of samples. In our experiments, we use a Gaussian kernel $K(x, y) = \exp\left(-\frac{\|x-y\|_2^2}{2\sigma^2}\right)$, with the bandwidth parameter $\sigma = 10$.

For experiments involving the `DMSB` method (Chen et al., 2023), we instead use an adaptive bandwidth $\sigma$ provided in the official implementation[3]. In this case, $\sigma$ is computed as the average squared pairwise distance between all distinct samples in the batch.

## C.2 TRAINING DETAILS

**Energy.** To ensure stability, we accumulate gradients across all time steps $k = 0, \ldots, K$, using mini-batch estimates $\widehat{\mathcal{J}}_\theta$ of the energy function $\mathcal{J}_\theta$ (see Eq. (12)). For each time step $t_k$, we sample a mini-batch $X_k = \{x_k^1, \ldots, x_k^B\} \sim \rho_k$, with a fixed batch size $B$. The energy is then estimated as:

$$\mathcal{J}_\theta(\rho_k) \approx \widehat{\mathcal{J}}_\theta(X_k) = \frac{1}{B} \sum_{i=0}^B \left[ V_{\theta_1}(x_k) + \frac{1}{B} \sum_{j=0}^B W_{\theta_2}(x_i - x_j) \right] - \theta_3 \widehat{\mathcal{H}}(X_k), \tag{25}$$

where $\widehat{\mathcal{H}}(X_k)$ denotes the estimated entropy of $\rho_k$ discussed in the following section.

**Map.** In practice, we use two strategies to parameterize each transport map $T_\varphi^k$: for large-scale tasks, we assign a separate network to each time step; for moderate-dimensional tasks, we encode the time index $k$ as an additional input, i.e., $T_\varphi^k(x) = T_\varphi(x, k)$, which helps prevent overfitting. This approach performs better than using a network without any time embedding. Thanks to `Optax` (DeepMind et al., 2020), the inference of all $T_\varphi^k$ can be performed in parallel, treating them as a single 'generator' step. We also experimented with initializing the maps during the early training epochs by aligning them with discrete OT maps computed using the `OTT-JAX` (Cuturi et al., 2022).

**Training.** We aggregate gradients for $\mathcal{J}_\theta$ and $T_\varphi^k$ across all time steps. We also experimented with alternative aggregation strategies, but found that equal aggregation across all time steps was the

---

[3] :octocat: `https://github.com/TianrongChen/DMSB/tree/main`

---

**Algorithm 1:** `iJKOnet` Training

---

**Input:**
    Sequence of measures $\{\rho_k\}_{k=0}^K$ (accessible via samples);
    Mapping networks $T_\varphi^k : \mathcal{X} \to \mathcal{X}$;
    Potential network $V_{\theta_1} : \mathcal{X} \to \mathbb{R}$, interaction network $W_{\theta_2} : \mathcal{X} \to \mathbb{R}$;
    Diffusion coefficient $\theta_3 \in \mathbb{R}^+$, time step $\tau$, max inner iterations $I_T$;
**Output:**
    Learned parameters $\theta = (\theta_1, \theta_2, \theta_3)$ and $\varphi$;
**while** not converged **do**
    **for** $i \leftarrow 1$ **to** $I_T$ **do**
        // Update transport maps $T_\varphi^k$
        **for** $k \leftarrow 0$ **to** $K - 1$ **do**
            Sample batch $X_k \sim \rho_k$, $X_{k+1} \sim \rho_{k+1}$;
            $X_{k+1}^{pred} \leftarrow T_\varphi^k(X_k)$;
            $\mathcal{L}_\varphi^k \leftarrow \mathcal{J}_\theta(X_{k+1}^{pred}) + \frac{1}{2\tau}\|X_{k+1}^{pred} - X_{k+1}\|_2^2$;
        $\varphi \leftarrow \varphi - \nabla_\varphi \sum_k \mathcal{L}_\varphi^k$;
    // Update energy parameters $\theta$
    **for** $k \leftarrow 0$ **to** $K - 1$ **do**
        Sample batch $X_k \sim \rho_k$;
        $X_{k+1}^{pred} \leftarrow T_\varphi^k(X_k)$;
        $\mathcal{L}_\theta^k \leftarrow -\mathcal{J}_\theta(X_{k+1}^{pred}) + \mathcal{J}_\theta(X_{k+1})$;
    $\theta \leftarrow \theta - \nabla_\theta \sum_k \mathcal{L}_\theta^k$;

---

most effective. During optimization, we perform multiple update steps for the transport maps $T_\varphi^k$, parameterized by $\varphi$, while updating the energy function $\mathcal{J}_\theta$, parameterized by $\theta$, only once. This follows standard practice in min–max optimization (Goodfellow et al., 2020). The overall procedure is summarized in the pseudocode in Algorithm 1.

**Stability.** The main challenge was avoiding suboptimal energy functionals that did not converge to zero. We found that the simplest setup (combined with careful hyperparameter tuning using the `Optuna` framework (Akiba et al., 2019)) was the most effective. We additionally experimented with common stabilization techniques from GAN training, including gradient penalties (Gulrajani et al., 2017), spectral normalization (Miyato et al., 2018; Jiang et al., 2018), and extragradient updates (Daskalakis et al., 2018). Overall, we perform optimization for single energy functional, this naturally acts as a regularizer, improving the stability of the training.

**Scalability.** Our method scales well with the number of time points. Leveraging `JAX` (Bradbury et al., 2018) and `Optax` (DeepMind et al., 2020), we avoid memory issues even for large batches and many time steps, since we do not backpropagate through time as in `JKOnet` (Bunne et al., 2022b). Each step is processed independently, and the loss is computed by averaging outputs from separate networks, avoiding large computation graphs. Training time on GPU is comparable to `JKOnet`*, but our approach eliminates the costly precomputation of OT couplings. In high-dimensional settings (50D–100D), `JKOnet`* exceeds GPU memory limits and requires CPU-based precomputations, making it substantially slower.

## C.3 ENTROPY ESTIMATION DETAILS

Prior to training, we estimated $\widehat{\mathcal{H}}(\rho_k)$ for each $k = 0, \ldots, K$ using the Kozachenko–Leonenko nearest-neighbor estimator (Kozachenko, 1987; Berrett et al., 2019) with 5 nearest neighbors. We used the publicly available implementation from (Butakov et al., 2024)[4]. As discussed in §3.3, we

---

[4]`https://github.com/VanessB/mutinfo`

use the following equation for estimating $\widehat{\mathcal{H}}(T_\varphi^k \sharp \rho_k)$:

$$\widehat{\mathcal{H}}(T_\varphi^k \sharp \rho_k) = \widehat{\mathcal{H}}(\rho_k) - \int_{\mathcal{X}} \log |\det \nabla_x T_\varphi^k(x)| \, \mathrm{d}\rho_k(x). \tag{26}$$

Since our map is not required to be invertible during training, as is the case for gradient-based ICNN (Amos et al., 2017) parameterizations (Bunne et al., 2022b), we compute the sign and the natural logarithm of the absolute value of the Jacobian determinant, both of which are supported by modern computational frameworks (Bradbury et al., 2018; Paszke, 2019).

## C.4 OPTIMIZER

We optimize two loss functions with respect to $\mathcal{J}_\theta$ and $T_\varphi^k$ using the Adam optimizer (Kingma, 2014). For $\mathcal{J}_\theta$, we use parameters $\beta_1 = 0.9$, $\beta_2 = 0.999$, $\varepsilon = 1 \times 10^{-8}$, and a constant learning rate of $5 \times 10^{-4}$, with gradient clipping applied using a maximum global norm of 10. For $T_\varphi^k$, we use parameters $\beta_1 = 0.5$, $\beta_2 = 0.9$, and $\varepsilon = 1 \times 10^{-8}$, with a learning rate of $1 \times 10^{-3}$. Training is performed with mini-batches of size 500.

## C.5 NETWORK ARCHITECTURE

The neural networks for the potential $V_{\theta_1}$ and interaction $W_{\theta_2}$ energies are multi-layer perceptrons (MLPs) with two hidden layers of size 64, using `softplus` activation functions, and a one-dimensional output layer. The neural network for the optimal transport maps $T_\varphi^k$ is also an MLP with two hidden layers of size 64. The time step $k$ is concatenated with the input $x_k^i$, and the network uses `selu` activations, with an output layer that matches the input dimension of $x_k^i$.

## C.6 HARDWARE

Experiments were conducted on a CentOS Linux 7 (Core) system with NVIDIA A100 GPUs. Most of the computation time was spent on metric evaluation. Leveraging JAX's just-in-time compilation, our method completed 100 training epochs in approximately one minute.

Table 7: Training time (in hours) for **EB** (Moon et al., 2019) dataset for 5 and 100 dimensions.

| **Solver** | **Dimension** | mean | std |
|---|---|---|---|
| iJKOnet$_V$ | 100 | 0.333 | 0.020 |
| iJKOnet$_{t,V}$ | 100 | 0.338 | 0.023 |
| iJKOnet$_V$ | 5 | 0.321 | 0.005 |
| iJKOnet$_{t,V}$ | 5 | 0.332 | 0.001 |

## C.7 FUNCTIONALS

For easier representation of our experimental results, we use abbreviations for each potential, each abbreviation being shown in parentheses and the definitions can be found in (Terpin et al., 2024, Appendix F): Wavy Plateau (**WP**), Double Exponential (**DE**), Rotational (**RO**), ReLU (**RE**), Flat (**FT**), Friedman (**FN**), Watershed (**WS**), Ishigami (**IG**), Flowers (**FL**), Bohachevsky (**BC**), Sphere (**SP**), Styblinski-Tang (**ST**), Oakley–Ohagan (**OO**), Zigzag Ridge (**ZR**), and Holder Table (**HT**).

# D  PROOFS

To begin with, we recall the notion of strong smoothness, see (Beck, 2017, §5.1). A functional $F : \mathcal{X} \to \mathbb{R}$ is called $\frac{1}{\beta}$ - *strongly smooth* if it is continuously differentiable on $\mathcal{X}$ and it holds:

$$\beta \|\nabla F(x) - \nabla F(y)\|_2 \leq \|x - y\|_2 \quad \forall x, y \in \mathcal{X}.$$

Now we proceed to some auxiliary results.

**Lemma D.1** (Solution to the JKO problem with potential energy is unique). *Let $\mathcal{J}^*(\rho) = \int_{\mathcal{X}} V^*(x)\mathrm{d}\rho(x)$, $\rho_0 \in \mathcal{P}_{2,ac}(\mathcal{X})$ and $\rho_1 = \mathrm{JKO}_{\tau\mathcal{J}^*}(\rho_0)$. Then $T_{V^*} = \arg\min_{T:\mathcal{X}\to\mathcal{X}} \mathcal{L}(V^*, T)$ is the (unique) optimal transport map between $\rho_0$ and $\rho_1$. In particular, $T_{V^*}\sharp\rho_0 = \rho_1$.*

*Proof.* Consider the functional $\rho \mapsto d^2_{\mathbb{W}_2}(\rho_0, \rho)$, where $\rho \in \mathcal{P}_2(\mathcal{X})$. In what follows, we prove that this functional is strictly convex.

Consider $\rho_a, \rho_b \in \mathcal{P}_2(\mathcal{X})$; $\alpha > 0$. Let $\rho = \alpha\rho_a + (1-\alpha)\rho_b$. Note that $\rho_0$ is absolutely continuous. By the Brenier's theorem, it holds:

$$d^2_{\mathbb{W}_2}(\rho_0, \rho_a) = \int \|x - T_a(x)\|_2^2 d\rho_0(x) \; ; \quad d^2_{\mathbb{W}_2}(\rho_0, \rho_b) = \int \|x - T_b(x)\|_2^2 d\rho_0(x)$$

for the corresponding (unique) OT maps $T_a : T_a\sharp\rho_0 = \rho_a$; $T_b : T_b\sharp\rho_0 = \rho_b$.

Now consider the weighted sum:

$$\alpha d^2_{\mathbb{W}_2}(\rho_0, \rho_a) + (1-\alpha)d^2_{\mathbb{W}_2}(\rho_0, \rho_b) = \int_{\mathcal{X}} \left(\alpha\|x - T_a(x)\|_2^2 + (1-\alpha)\|x - T_b(x)\|_2^2\right) d\rho_0(x)$$

$$= \int_{\mathcal{X}} \|x - y\|_2^2 d\pi_{ab}(x, y),$$

where plan $\pi_{ab} \in \Pi(\rho_0, \rho)$ has the following conditionals:

$$\pi(y|x) = \alpha\delta(y = T_a(x)) + (1-\alpha)\delta(y = T_b(x)).$$

Since the OT map between $\rho_0$ and $\rho$ is unique (and deterministic), we have

$$d^2_{\mathbb{W}_2}(\rho_0, \rho) = \min_{T\sharp\rho_0=\rho} \int_{\mathcal{X}} \|x - T(x)\|_2^2 d\rho(x) < \int_{\mathcal{X}} \|x - y\|_2^2 d\pi_{ab}(x, y),$$

which yields the strict convexity of $\rho \mapsto d^2_{\mathbb{W}_2}(\rho_0, \rho)$.

The main statement of the Lemma follows from the strict convexity of $\rho \mapsto \int_{\mathcal{X}} V^*(x)d\rho(x) + d^2_{\mathbb{W}_2}(\rho_0, \rho)$ and Brenier's theorem. $\square$

Now we are ready to prove our main quality bound theorem.

*Proof.* (Theorem 3.1).

*Preliminary Note:* The facts from convex analysis (properties of convex functions and their conjugates) which we use below could be found in (Kornilov et al., 2024, Lemma 1).

At first, we simplify the expression for the JKO loss:

$$\mathcal{L}(V, T_V) = \int V(T_V(x))d\rho_0(x) - \int V(y)d\rho_1(y) + \frac{1}{2\tau}\int \|x - T_V(x)\|_2^2 d\rho_0(x)$$

$$= \frac{1}{\tau}\int V_q(T_V(x))d\rho_0(x) - \frac{1}{\tau}\int V_q(y)d\rho_1(y) - \frac{1}{\tau}\int \langle x, T_V(x)\rangle d\rho_0(x) \qquad (27)$$

$$+ \underbrace{\int \frac{\|y\|_2^2}{2\tau}d\rho_1(y) + \int \frac{\|x\|_2^2}{2\tau}d\rho_0(x)}_{\overset{\text{def}}{=}C(\rho_0,\rho_1)},$$

where $V_q = \tau V + \frac{1}{2}\|\cdot\|_2^2$.

From (27) we note that $T_V = \arg\min_T \int \left(V_q(T_V(x)) - \langle x, T_V(x)\rangle\right)d\rho_0(x)$. From the convex analysis, it follows that $T_V(x) = \nabla\overline{V_q}(x)$ delivers the minimum; $\overline{V_q}$ is the (Fenchel) conjugate of $V_q$. Note that $T_V$ is measurable since it is continuous almost surely (Rockafellar, 1970, Theorem 25.5). Substituting $\nabla\overline{V_q}$ into (27) on par with Fenchel-Young equality yields:

$$\mathcal{L}(V, T_V) = -\frac{1}{\tau}\int \overline{V}_q(x)d\rho_0(x) - \frac{1}{\tau}\int V_q(y)d\rho_1(y) + C(\rho_0, \rho_1). \qquad (28)$$

Note that $T_{V^*} \sharp \rho_0 = \rho_1$ (Lemma D.1). Therefore, eq. (27) for $\mathcal{L}(V^*, T_{V^*})$ could be simplified:

$$\mathcal{L}(V^*, T_{V^*}) =$$

$$\frac{1}{\tau} \underbrace{\int V_q^*(T_{V^*}(x)) d\rho_0(x)}_{=\int V_q^*(y) d\rho_1(y)} - \frac{1}{\tau} \int V_q^*(y) d\rho_1(y) - \frac{1}{\tau} \int \langle x, T_{V^*}(x) \rangle d\rho_0(x) + C(\rho_0, \rho_1) =$$

$$-\frac{1}{\tau} \int \langle x, T_{V^*}(x) \rangle d\rho_0(x) + C(\rho_0, \rho_1). \quad (29)$$

Now we analyze the gap $\varepsilon(V)$ between optimal and optimized JKO losses. Leveraging (28) and (29) yields:

$$\tau \varepsilon(V) = \tau \mathcal{L}(V^*, T_{V^*}) - \tau \mathcal{L}(V, T_V) = \int \overline{V_q}(x) d\rho_0(x) + \int V_q(y) d\rho_1(y) - \int \langle x, T_{V^*}(x) \rangle d\rho_0(x). \quad (30)$$

Now we note that $T_{V^*} = \nabla \overline{V_q^*} : \mathcal{X} \to \mathbb{R}^D$. From the properties of convex functions it follows that $\nabla V_q^* : \mathcal{X} \to \mathbb{R}^D$ defines the *inverse* Optimal Transport mapping. In particular (we assume $V_q^*$ to be convex) $\nabla V_q^* \sharp \rho_1 = \rho_0$. Then, changing the variables in (30) results in:

$$\tau \varepsilon(V) = \int \overline{V_q}(\nabla V_q^*(y)) d\rho_1(y) + \int V_q(y) d\rho_1(y) - \int \langle \nabla V_q^*(y), y \rangle d\rho_1(y)$$

$$= \int_{\mathcal{X}} \left[ \overline{V_q}(\nabla V_q^*(y)) + V_q(y) - \langle \nabla V_q^*(y), y \rangle \right] d\rho_1(y). \quad (31)$$

$$= \int_{\mathcal{X}} \mathcal{D}_{\overline{V_q}}(\nabla V_q^*(y), \nabla V_q(y)) d\rho_1(y), \quad (32)$$

where $\mathcal{D}_{\overline{V_q}}(\cdot, \cdot)$ is the *Bregman* divergence, see (Banerjee et al., 2005, Def. 1) for the definition and (Banerjee et al., 2005, Appendix A) for the "Dual Divergences" property used in the transition from (31) to (32).

Since $V_q$ is $\frac{1}{\beta}$-smooth, then $\overline{V_q}$ is $\beta$-strongly convex. Therefore, by the property of strongly convex functions (we pick $y \in \text{supp}(\rho_1)$):

$$\frac{\beta}{2} \|\nabla V_q^*(y) - \nabla V_q(y)\|_2^2 \leq \overline{V_q}(\nabla V_q^*(y)) - \overline{V_q}(\nabla V_q(y)) - \langle \nabla \overline{V_q}(\nabla V_q(y)), \nabla V_q^*(y) - \nabla V_q(y) \rangle$$

$$= \mathcal{D}_{\overline{V_q}}(\nabla V_q^*(y), \nabla V_q(y)). \quad (33)$$

Combining (32) and (33) yields:

$$\tau \varepsilon(V) \geq \frac{\beta}{2} \int_{\mathcal{X}} \|\nabla V_q^*(y) - \nabla V_q(y)\|_2^2 d\rho_1(y).$$

To complete the proof, we are left to note that $\nabla V_q^*(y) - \nabla V_q(y) = \tau \big( \nabla V^*(y) - \nabla V(y) \big)$. $\quad \square$

