# OpenReview forum: "Learning of Population Dynamics: Inverse Optimization Meets JKO Scheme"
_ICLR.cc/2026/Conference — ICLR 2026 Poster_

### Official Review · Reviewer_vGio · 2025-10-28

**Soundness:** 3
**Presentation:** 3
**Contribution:** 2
**Rating:** 2
**Confidence:** 4

**Summary:**

The paper is an attempt to use the (famous) JKO optimization scheme to learn a dynamical system from a time- series $\rho_1, \cdots, \rho_M$  where $\rho_i$ represent the data of the system at time $T_1< T_2 < \cdots < T_M$.  The JKO scheme are expressed as proximal operator of the $W_2^2$ optimal cost with a functional (typically of the form energy plus entropy plus possibly interaction terms described by a kernel).   The approach taken here is to parametrize a transport map (for the $W^2$ term) and to parametrize  the energy terms using neural architectures and to estimate the entropy terms via an entropy estimator.

**Strengths:**

The JKO formulation of gradient flow as a proximal optimzation problem is a fundamental tool in probbaility theory.  It has been found difficult to use it effectively as a generating tool and and the reformulation of the JKO functional in this paper  is interesting and could be promising.

**Weaknesses:**

1) The main weakness is that the paper does not fully implement the proposed loss functional. Indeed the the interaction terms and the entropy terms are ignored and the same architecture as in JKONet is used.  This reduces the class of PDEs considered to a very small class of linear PDE. The mismatch between the theory which considers general PDEs and the implementation is too great to make the paper convincing.   Ignoring the entropy term,  which I suspect is very difficult to estimate,  is a serious defect of the approach.

2) The proposed architecture requires the construction of a transport map which is both expensive and only works theoretically when the measures have density.  The authors propose to replace input convex neural networks (since the transport maps are gradient of a convex functions) by a more general architecture but I do not see why this will make the problem more scalable or stable.  Maybe a approach using triangular map as in the Yousef Mazrouk group at MIT may be more scalable?

**Questions:**

1) Can you clarify what are the obstacles to implement an interaction kernel and the entropy term in the algorithm?

2) Can you comment on the difficulty of computing the transport maps?

---

> ### Author Response · Authors · 2025-11-23
> **Official Comment by Authors. Part 1**
>
> # Response to Reviewer vGio
>
> Dear Reviewer vGio,
>
> We thank you the reviewer for recognizing our reformulation of the JKO functional that could be promising.
>
> ---
> **1. "The main weakness is that the paper does not fully implement the proposed loss functional. Indeed the the interaction terms and the entropy terms are ignored and the same architecture as in JKONet is used. [...] The mismatch between the theory which considers general PDEs and the implementation is too great to make the paper convincing"**
>
> We respectfully disagree with the claim that we do not fully implement the proposed loss functional. As stated in Section 5, and as demonstrated in Appendix B.1.2 (interaction energy) and Appendix B.1.3 (internal/entropy energy), we _did_ implement these additional terms and carefully evaluated them. Unfortunately, these energies cannot be reliably recovered in practice, and for this reason the main text focuses only on the potential term. To avoid any ambiguity, for our 5D single-cell experiment (Section 5.2 of the main text) the revised manuscript now includes extended **Table 1** (also provided below), summarizing _all_ combinations of energy parametrizations and illustrating that interaction and internal terms often degrade convergence—consistent with our findings in the appendix.
>
> In addition, Appendix B.1.2 now includes an extra experiment in the **paired** setup for interaction energy. Figure 5 shows that even in this favorable 2D scenario, neither our method nor $\texttt{JKOnet}^\ast$ [1] succeeds in accurately recovering the interaction energy. This should be contrasted with the **unpaired** case in Figure 6, where none of the energies can be recovered. These results reinforce our conclusion that the difficulty is not specific to our approach but is intrinsic to both our parametrization and that of $\texttt{JKOnet}^\ast$. Despite claims in [1] that the diffusion term can be recovered, Appendix B.1.3 clearly demonstrates that it converges to incorrect values in controlled experiments.
>
> We have made this motivation more explicit in the revised manuscript: although we implemented the full functional, current estimators for interaction and internal energies remain unreliable, and this limitation affects both our method and $\texttt{JKOnet}^\ast$. This explains why the main text focuses on the potential energy, while the appendix provides the complete analysis.
>
>
> | Method  | $t_2$  | $t_4$ |
> | -------------------------- | ---------------- | ---------------- |
> | **Baselines**  |  |  |
> | Vanilla-SB | 1.49 ± 0.063  | 1.55 ± 0.034  |
> | DMSB | 1.13 ± 0.082   | 1.45 ± 0.16 |
> | TrajectoryNet | 2.03 ± 0.04  | 1.93 ± 0.08 |
> | MMSB| 1.27 ± 0.028  | 1.57 ± 0.048     |
> | **Static**|   |  |
> | JKOnet$^\ast_V$  | 1.145 ± 0.033    | 2.529 ± 0.014    |
> | JKOnet$^\ast_{V+U}$ | 1.099 ± 0.119    | 2.537 ± 0.054    |
> | JKOnet$^\ast_{V+W}$| 1.419 ± 0.173    | 2.510 ± 0.094    |
> | JKOnet$^\ast_{W+U}$| 1.887 ± 0.017    | 1.739 ± 0.037  |
> | JKOnet$^\ast$  | 1.361 ± 0.257    | 2.557 ± 0.042    |
> | **Static (Ours)**|  |  |
> | iJKOnet$_V$ | 1.082 ± 0.011  | 1.147 ± 0.001 |
> | iJKOnet$_{V+U}$ | *1.065 ± 0.018* | *1.150 ± 0.004* |
> | iJKOnet$_{V+W}$ | 2.865 ± 0.166    | 1.376 ± 0.015 |
> | iJKOnet$_{W+U}$| 1.649 ± 0.005    | 0.868 ± 0.005  |
> | iJKOnet      | 3.577 ± 0.166    | 1.395 ± 0.032  |
> | **Time-varying**  |  |  |
> | JKOnet*$_{t,V}$  | 4.414 ± 1.499 | 2.771 ± 0.197 |
> | iJKOnet$_{t,V}$ **(Ours)**  | **0.983 ± 0.037** | **0.849 ± 0.021** |
>
> **2. Ignoring the entropy term, which I suspect is very difficult to estimate, is a serious defect of the approach. [...] Can you clarify what are the obstacles to implement an interaction kernel and the entropy term in the algorithm?**
>
> As discussed in Appendix B.1.3, the challenges with the entropy term primarily stem from the difficulty of estimating entropy reliably in moderate or high dimensions. As discussed in the main text, diffusion-coefficient learning becomes unstable because existing entropy estimators degrade rapidly with dimension. While [1] reports results only in 5D, our synthetic experiments show that even in 10D the estimates become inconsistent: both Gaussian mixture–based density estimators (used by default in [1]) and the Kozachenko–Leonenko nearest-neighbor estimator [3] fail, and in 100D the estimator returns NaNs. These limitations are inherent to current entropy-estimation techniques and orthogonal to our method; developing more robust estimators remains an open problem.
>
> As discussed in Appendix B.1.2, for the interaction energy, the main obstacle is the Monte-Carlo approximation of the required double integral. We experimented with the same RBF-kernel parameterization proposed in [4], but this inductive bias proved even less reliable and less stable than our simple MLP-based parameterization. Thus, the difficulty is not architectural but stems from fundamental variance and bias issues in estimating interaction energies from samples.

---

> > ### Author Response · Authors · 2025-11-23
> > **Official Comment by Authors. Part 2**
> >
> > **3. "The proposed architecture requires the construction of a transport map which is both expensive and only works theoretically when the measures have density. [...] Can you comment on the difficulty of computing the transport maps?"**
> >
> > It is indeed true that theoretical guarantees for transport maps are stated for measures with density, but this is a common assumption in practice and holds for most applications of interest. In our framework, each transport map is represented as a neural network, which allows the inference to be performed efficiently. Moreover, using JAX, we can parallelize computations across multiple maps, including one per time step, which significantly reduces computational cost and makes the approach scalable. As a result, while the construction of transport maps may appear expensive in principle, in practice it is efficiently implemented and fully practical for the scenarios we consider. Following your question, We added more details regarding map evaluation and scalability in Appendix C.2.
> >
> > ---
> > **4. "The authors propose to replace input convex neural networks (since the transport maps are gradient of a convex functions) by a more general architecture but I do not see why this will make the problem more scalable or stable. Maybe a approach using triangular map as in the Yousef Mazrouk group at MIT may be more scalable? [...]"**
> >
> > We appreciate the reviewer’s suggestion regarding triangular maps. In practice, we found that input-convex architectures introduce substantial computational and optimization challenges. As reported in Table 1 of [1], replacing ICNNs with vanilla MLPs along with Monge-gap regularization [5] (as in variants of JKOnet [2]) still requires solving an optimal transport problem at every inner iteration, leading to prohibitive slowdowns and making large-scale training impractical.
> >
> > Our approach instead uses unconstrained MLPs to parameterize transport maps. While the true optimal maps are gradients of convex functions, enforcing architectural convexity constraints (such as in ICNNs) complicates training and may limit expressiveness. Standard MLPs, in contrast, train efficiently, avoid these structural restrictions, and can be parallelized across time steps using JAX, which substantially improves scalability. For 2D examples, maps learned by MLPs achieve similar or better performance compared to ICNNs, confirming the validity of this approach. In larger experiments, ICNNs consistently failed to scale, whereas our simple MLP parametrization remains stable and effective.
> >
> > Exploring triangular or other structured map classes is indeed an interesting direction, and we consider this a promising avenue for future work.
> >
> > ---
> > **Conclusion**
> >
> > We hope these clarifications address your concerns. Should you have additional questions, we would be happy to provide further details. Otherwise, we kindly ask you to consider raising your evaluation score in light of these explanations.
> >
> > ---
> > **References**
> >
> > [1] Terpin, Antonio, et al. "Learning diffusion at lightspeed." _Advances in Neural Information Processing Systems_ 37 (2024): 6797-6832.
> >
> > [2] Bunne, Charlotte, et al. "Proximal optimal transport modeling of population dynamics." _International Conference on Artificial Intelligence and Statistics_. PMLR, 2022.
> >
> > [3] Berrett, Thomas B., Richard J. Samworth, and Ming Yuan. "Efficient multivariate entropy estimation via k-nearest neighbour distances." (2019): 288-318.
> >
> > [4] Zhang, Zhenyi, et al. "Modeling Cell Dynamics and Interactions with Unbalanced Mean Field Schr\" odinger Bridge." arXiv preprint arXiv:2505.11197 (2025).
> >
> > [5] Uscidda, Théo, and Marco Cuturi. "The monge gap: A regularizer to learn all transport maps." International Conference on Machine Learning. PMLR, 2023.

---

### Official Review · Reviewer_tCBu · 2025-11-01

**Soundness:** 3
**Presentation:** 4
**Contribution:** 3
**Rating:** 6
**Confidence:** 3

**Summary:**

In this paper, the author propose  a method to learn the population dynamics, which combine JKO framework with inverse optimization techniques. In addition, this method relays on adversarial training procedure. Theoretical guarantees are also provided in this paper.

**Strengths:**

1. This paper is well written.

2. The figures are well-prepared and greatly facilitate the understanding of the content.

3. The theoretical proofs is provided, which give solid theoretical guarantees of the proposed method.

4. The background section is clearly presented and offers helpful context.

**Weaknesses:**

1. The interchange of the min and Σ operators between Equations (10) and (11) lacks justification or analysis. It is unclear under what conditions this exchange is mathematically valid.

2. The method relies on an adversarial training procedure, which is known to introduce instability. It would be important to discuss whether any measures were taken to mitigate this issue.

3. The work builds heavily on existing studies (e.g., Terpin et al., 2024; JKOnet∗). The specific contributions and novelty of the present approach should be more clearly emphasized.

4. Several relevant references appear to be missing from the related work section, such as:

[1] Zhang Z, et al. Modeling Cell Dynamics and Interactions with Unbalanced Mean Field Schr\" odinger Bridge[J]. arXiv preprint arXiv:2505.11197, 2025.

[2] Li R, et al. WeightFlow: Learning Stochastic Dynamics via Evolving Weight of Neural Network[J]. arXiv preprint arXiv:2508.00451, 2025.

[3] Zhang Y, Levin M. Equilibrium flow: From Snapshots to Dynamics[J]. arXiv preprint arXiv:2509.17990, 2025.

**Questions:**

- Could you explain about the potential stability of the proposed method resulted by the adversarial training procedure, and what further improvements could be made?

- Could you more explicitly summarize the key algorithmic or theoretical innovations that distinguish your approach from its predecessors?

---

> ### Author Response · Authors · 2025-11-23
>
> Dear Reviewer tCBu,
>
> Thank you for your positive feedback. We appreciate your comments on the quality of the writing, the clarity of the background section, and the helpfulness of the figures. We are also grateful for your acknowledgement of the theoretical results and proofs, and we are glad that they contributed to the overall strength of the paper.
>
> ---
> **1. "The interchange of the min and Σ operators between Equations (10) and (11) lacks justification or analysis. It is unclear under what conditions this exchange is mathematically valid."**
>
> Thank you for pointing this out. The interchange is mathematically valid because the minimization is taken over an independent finite sum of per-snapshot terms, which allows the min and the summation to be exchanged. In the revised manuscript, we have clarified this step and expanded the explanation, please see the updates in section 3.2. We also provide additional context regarding the standard trick in the JKO literature [1] involving the exchange of $\rho^k$ and $T^k$.
>
> ---
> **2. "The method relies on an adversarial training procedure, which is known to introduce instability. It would be important to discuss whether any measures were taken to mitigate this issue. [...] Could you explain about the potential stability of the proposed method resulted by the adversarial training procedure, and what further improvements could be made?"**
>
> Thank you for this question! In fact, we experimented with several setups to address potential instability in the adversarial training. In the revised manuscript (Appendix C.2), we have clarified the different approaches we explored, including extragradient updates, gradient penalties, alternative aggregation of loss terms, spectral normalization of the Jacobian, and guidance using discrete optimal maps. In general, since we perform optimization for single energy functional, this naturally acts as a regularizer, improving the stability of the training. The main challenge was avoiding suboptimal energy functionals that did not converge to zero. We found that the simplest setup (combined with careful hyperparameter tuning using the $\texttt{Optuna}$ framework [2]) was the most effective.
>
> ---
> **3. "The work builds heavily on existing studies (e.g., Terpin et al., 2024; JKOnet∗). The specific contributions and novelty of the present approach should be more clearly emphasized. [...] Could you more explicitly summarize the key algorithmic or theoretical innovations that distinguish your approach from its predecessors?"**
>
> We respectfully note that Section 4.2 is dedicated to outlining the advantages of our method over prior work, and we have expanded this discussion in response to your comments. Briefly, our method inherits from $\texttt{JKOnet}$ [1] the use of parameterized models for both the energy functional $\mathcal{J}\_\theta$ and the transport map $T_\varphi$, which avoids reliance on precomputed discrete optimal transport plans. This allows for more flexible mappings between $\rho_k$ and $\rho_{k+1}$. From $\texttt{JKOnet}^\ast$ [3], we adopt a more expressive formulation of the energy functional. Unlike $\texttt{JKOnet}^\ast$, our formulation does not rely on analytical expressions of the JKO step (8) optimality conditions; instead, it solves the step directly via the inner minimization in (11). This approach makes the method more readily extendable to general energy forms, such as porous medium energies [4], However, exploring such extensions is beyond the scope of the current work.
>
> **4. "Several relevant references appear to be missing from the related work section [...]."**
>
> Thank you for pointing out these references. We have included them in the revised manuscript. We note that the first paper was already cited in the original submission, and the third work appeared on arXiv only three days before the full-text submission deadline. The Wasserstein gradient flow literature is very broad, making complete coverage challenging, but we have made a substantial effort to provide a comprehensive overview in Appendix A.1.1, which spans two pages.
>
> ---
> **Conclusion**
>
> We hope these clarifications address your concerns. Should you have additional questions, we would be happy to provide further details. Otherwise, we kindly ask you to consider raising your evaluation score in light of these explanations.
>
> ---
> **References**
>
> [1] Bunne, Charlotte, et al. "Proximal optimal transport modeling of population dynamics." AISTATS, 2022.
>
> [2] Akiba, Takuya, et al. "Optuna: A next-generation hyperparameter optimization framework." KDD, 2019.
>
> [3] Terpin, Antonio, et al. "Learning diffusion at lightspeed." NeurIPS, 2024.
>
> [4] Alvarez-Melis, David, Yair Schiff, and Youssef Mroueh. "Optimizing Functionals on the Space of Probabilities with Input Convex Neural Networks." (2022).

---

> > ### Comment · Reviewer_tCBu · 2025-11-26
> >
> > Thank the author for the detailed response, which addresses all of my concerns. I have therefore increased my score.

---

> > > ### Author Response · Authors · 2025-11-27
> > >
> > > Dear Reviewer tCBu,
> > >
> > > We sincerely appreciate your thoughtful feedback and are grateful for the updated score. We remain happy to address any further questions you may have.

---

### Official Review · Reviewer_XRFW · 2025-11-07

**Soundness:** 3
**Presentation:** 3
**Contribution:** 3
**Rating:** 6
**Confidence:** 3

**Summary:**

This paper introduces iJKOnet, an approach to learning population dynamics by bridging inverse optimization techniques with the Jordan-Kinderlehrer-Otto (JKO) scheme. The method frames the task as a min-max optimization problem for identifying the underlying energy functional that governs observed distributions, enabling end-to-end adversarial training without restrictive neural architectures. Theoretical guarantees are provided for recovery of potential-driven dynamics, and empirical results on both synthetic and real datasets show improved performance over prior JKO-based baselines.

**Strengths:**

- iJKOnet is different from previous methods that either require potential-only energies or precomputed OT couplings. The formulation of energy functional recovery using inverse optimization within the JKO scheme provides a conceptually clear route to modeling population level dynamics from discrete snapshots, directly leveraging optimal transport geometry.
- Theorem 3.1 gives a non-trivial quality guarantee, explicitly bounding the distance between the gradients of learned and ground-truth potential functions in terms of the inverse optimization gap. This is a concrete advance compared to many contemporary works that lack precise recovery guarantees.
- iJKOnet dispenses with input-convex neural networks for parameterizing transport maps, allowing the use of more scalable and widely applicable neural network architectures such as MLPs and ResNets

**Weaknesses:**

- The main theoretical guarantee (Theorem 3.1) exclusively addresses potential energy functionals with strongly convex smooth potentials, while the practical method is claimed to work for broader energy functionals (e.g., including interaction and internal terms). This leaves a theoretical gap between what the analysis covers and what is demonstrated empirically. Specifically, the inability to provide quality bounds for non-potential and more general functionals limits the rigor of the claims. (This could explain why the performance of iJKOnet is much worse than baseline in Table 2)
- While iJKOnet is, in principle, applicable to full free-energy functionals, all main experiments revert to only learning potential energy, explicitly setting interaction and entropy terms to zero. As a result, the empirical support for learning richer dynamics is missing, and the scalability to very high-dimensional or truly multimodal interaction-dependent data is unproven.
- Some recent and directly related works [1,2,3,4] on similar task with unknown parameters, alternative population-level diffusion modeling, and advanced regularization techniques are not discussed.

ref:

[1] Computational and Statistical Asymptotic Analysis of the JKO Scheme for Iterative Algorithms to update distributions

[2] An Eulerian approach to regularized JKO scheme with low-rank tensor decompositions for Bayesian inversion

[3] WeightFlow: Learning Stochastic Dynamics via Evolving Weight of Neural Network

[4] Modeling Cell Dynamics and Interactions with Unbalanced Mean Field Schrodinger Bridge

**Questions:**

- In which regimes or datasets is JKO-based modeling justified or superior to flow-matching, neural ODEs, or Schrödinger bridge-based methods?
- How sensitive is iJKOnet to missing, irregularly spaced, or noisy population snapshots? Do learned energy functionals overfit to sampling artifacts in sparse settings?

---

> ### Author Response · Authors · 2025-11-23
> **Official Comment by Authors. Part 1**
>
> Dear Reviewer XRFW,
>
> We sincerely thank you for recognition of our approach, including the inverse-optimization formulation within the JKO scheme, the theoretical guarantees on potential recovery, and the flexible neural network parameterization of transport maps.
>
> ---
> **1. "The main theoretical guarantee (Theorem 3.1) exclusively addresses potential energy functionals with strongly convex smooth potentials, while the practical method is claimed to work for broader energy functionals (e.g., including interaction and internal terms). This leaves a theoretical gap between what the analysis covers and what is demonstrated empirically. Specifically, the inability to provide quality bounds for non-potential and more general functionals limits the rigor of the claims. [...]"**
>
> We acknowledge the reviewer’s concern that the absence of theoretical guarantees for interaction and internal energies may appear as a limitation. However, unlike the potential-energy case, the primary difficulty for the interaction term arises from the convolution with the unknown density when solving interaction-induced JKO step $\hat{\rho}_1 = \mathrm{JKO}_{\tau \mathcal{W}}(\rho_0)$, which makes a step-by-step extension of the potential-energy proof nontrivial.
>
> Importantly, our experiments in Appendix B.1.2 and B.1.3 show that, in practice, learning interaction and internal energies remains challenging: neither our method nor $\texttt{JKOnet}^\ast$ reliably recovers these terms. This suggests that, while theoretical bounds would be valuable, the practical limitations of learning these energies dominate the observed behavior. For details, see discussion below.
>
> ---
> **2. "While iJKOnet is, in principle, applicable to full free-energy functionals, all main experiments revert to only learning potential energy, explicitly setting interaction and entropy terms to zero. As a result, the empirical support for learning richer dynamics is missing, and the scalability to very high-dimensional or truly multimodal interaction-dependent data is unproven."**
>
> We would like to clarify that our focus on potential-energy parametrizations in the main text is a direct consequence of the empirical observations reported in Appendix B.1.2 (interaction energy) and Appendix B.1.3 (internal energy). In the revised version, we now include all combinations of parametrizations in Table 1 (also provided below for convenience) for 5D single-cell experiment (section 5.2), demonstrating that adding interaction or internal terms often disrupts the overall convergence of the algorithm for both our method and $\texttt{JKOnet}^\ast$.
>
> To further strengthen this point, Appendix B.1.2 now contains an additional **paired** experiment for interaction energy. As shown in Figure 5, even in this favorable 2D paired setup, neither our method nor $\texttt{JKOnet}^\ast$ is able to recover the all the interaction potentials accurately. This stands in contrast to the **unpaired** setting in Figure 6, where recovery fails entirely. These results indicate that the difficulty is not specific to our formulation but is shared across parametrizations. A similar issue arises for internal energy: in Figure 7 (Appendix B.1.3), both approaches converge to a parameter $\beta$, but further checks show that this value is incorrect, despite previous claims in [1].
>
> Because these challenges become even more severe in higher-dimensional or multimodal settings, we restricted our main experiments to potential-energy parametrizations. This choice does not reflect a limitation of $\texttt{iJKOnet}$ itself. Indeed, the method is fully compatible with the full free-energy formulation—but rather highlights an empirical limitation affecting all existing methods. Our results therefore provide evidence that learning richer interaction- and entropy-driven dynamics remains an open practical problem, rather than one that our approach uniquely fails to address.
>
> | Method  | $t_2$  | $t_4$ |
> | -------------------------- | ---------------- | ---------------- |
> | **Baselines**  |  |  |
> | Vanilla-SB | 1.49 ± 0.063  | 1.55 ± 0.034  |
> | DMSB | 1.13 ± 0.082   | 1.45 ± 0.16 |
> | TrajectoryNet | 2.03 ± 0.04  | 1.93 ± 0.08 |
> | MMSB| 1.27 ± 0.028  | 1.57 ± 0.048     |
> | **Static**|   |  |
> | JKOnet$^\ast_V$  | 1.145 ± 0.033    | 2.529 ± 0.014    |
> | JKOnet$^\ast_{V+U}$ | 1.099 ± 0.119    | 2.537 ± 0.054    |
> | JKOnet$^\ast_{V+W}$| 1.419 ± 0.173    | 2.510 ± 0.094    |
> | JKOnet$^\ast_{W+U}$| 1.887 ± 0.017    | 1.739 ± 0.037  |
> | JKOnet$^\ast$  | 1.361 ± 0.257    | 2.557 ± 0.042    |
> | **Static (Ours)**|  |  |
> | iJKOnet$_V$ | 1.082 ± 0.011  | 1.147 ± 0.001 |
> | iJKOnet$_{V+U}$ | *1.065 ± 0.018* | *1.150 ± 0.004* |
> | iJKOnet$_{V+W}$ | 2.865 ± 0.166    | 1.376 ± 0.015 |
> | iJKOnet$_{W+U}$| 1.649 ± 0.005    | 0.868 ± 0.005  |
> | iJKOnet      | 3.577 ± 0.166    | 1.395 ± 0.032  |
> | **Time-varying**  |  |  |
> | JKOnet*$_{t,V}$  | 4.414 ± 1.499 | 2.771 ± 0.197 |
> | iJKOnet$_{t,V}$ **(Ours)**  | **0.983 ± 0.037** | **0.849 ± 0.021** |

---

> > ### Author Response · Authors · 2025-11-23
> > **Official Comment by Authors. Part 2**
> >
> > **3. "Some recent and directly related works [1,2,3,4] on similar task with unknown parameters, alternative population-level diffusion modeling, and advanced regularization techniques are not discussed."**
> >
> > Thank you for pointing out these references. We have included them in the revised manuscript. We note that the forth paper was already cited in the original submission. The Wasserstein gradient flow literature is very broad, making complete coverage challenging, but we have made a substantial effort to provide a comprehensive overview in Appendix A.1.1, which spans two pages.
> >
> > ---
> > **4. "In which regimes or datasets is JKO-based modeling justified or superior to flow-matching, neural ODEs, or Schrödinger bridge-based methods?"**
> >
> > The strength of JKO-based modeling lies in settings where the underlying dynamics are naturally described by Wasserstein Gradient Flows (WGFs), particularly in physics-informed or energy-driven systems. In these regimes, our approach provides a principled alternative to flow matching (FM), neural ODEs, or Schrödinger bridge (SB) methods by directly parameterizing the energy landscape rather than the velocity or score.
> >
> > Empirically, Table 5 (Appendix B.2) shows that our method outperforms both FM and SB baselines. Furthermore, recent work [2] provides theoretical justification for using time-varying free-energy functionals — an aspect that was used in [1] and not explained. This suggests that our framework could be extended to fully time-dependent potentials in future work. Additionally, [3] argues that diffusion training is more naturally interpreted as flow matching to the velocity field of a WGF rather than as score estimation for a reverse SDE, suggesting that JKO-based methods may have broader relevance to generative modeling.
> >
> > Taken together, these results indicate that JKO-based modeling is particularly well-suited for systems governed by energy minimization principles and may become increasingly relevant in areas where diffusion processes and gradient-flow structures intersect.
> >
> > ---
> > **5. "How sensitive is iJKOnet to missing, irregularly spaced, or noisy population snapshots? Do learned energy functionals overfit to sampling artifacts in sparse settings?"**
> >
> > We may have misunderstood part of the question, so clarification would be appreciated. However, we note that the unpaired setting already corresponds to a missing and irregularly spaced regime, since individual samples are not temporally aligned. Following your suggestion, in Appendix A.3 we provide an ablation and detailed discussion of the paired vs. unpaired cases. As shown in Figure 3, our method is more robust in the unpaired (i.e., sparse and irregular) scenario than the approach of [1], suggesting that iJKOnet does not overfit to sampling artifacts even when observations are limited or irregular.
> >
> > ---
> > **Conclusion**
> >
> > We hope these clarifications address your concerns. Should you have additional questions, we would be happy to provide further details. Otherwise, we kindly ask you to consider raising your evaluation score in light of these explanations.
> >
> > ---
> > **References**
> >
> > [1] Terpin, Antonio, et al. "Learning diffusion at lightspeed." _Advances in Neural Information Processing Systems_ 37 (2024): 6797-6832.
> >
> > [2] Ferreira, Lucas CF, and Julio C. Valencia-Guevara. "Gradient flows of time-dependent functionals in metric spaces and applications to PDEs." Monatshefte für Mathematik 185.2 (2018): 231-268.
> >
> > [3] Vuong, An B., et al. "Are We Really Learning the Score Function? Reinterpreting Diffusion Models Through Wasserstein Gradient Flow Matching." arXiv preprint arXiv:2509.00336 (2025).

---

> ### Comment · Reviewer_XRFW · 2025-11-24
>
> I particularly appreciate the clarifications regarding the theoretical and practical challenges of learning interaction and internal energies. The additional experiments demonstrating that this is a shared limitation across existing methods are convincing.
>
> Furthermore, the discussion comparing JKO-based modeling to Flow Matching and Neural ODEs helps clearly contextualize the paper's contribution within the broader landscape of generative modeling and physics-informed learning.
>
> While I initially raised concerns about the gap between the theoretical analysis and the general functionals used in practice, I acknowledge the authors' effort to rigorously evaluate the method and the significance of the proposed iJKOnet framework.
>
> In light of the authors' clarifications and the overall quality of the revised paper, I am raising my rating to 8. However, to accurately reflect my level of expertise regarding the specific intricacies of JKO schemes and optimal transport geometry compared to the core technical depth of this work, I am lowering my confidence score to 2.

---

> > ### Author Response · Authors · 2025-11-24
> >
> > Dear  Reviewer XRFW,
> >
> > We sincerely thank you for raising the score and for the thoughtful feedback. We appreciate the recognition of our experiments and clarifications, and we remain happy to address any remaining questions to further support confidence in the technical correctness and depth of our work.

---

### Author Response · Authors · 2025-11-23
**General Response and Revision**

We sincerely thank all reviewers for their thoughtful feedback and for highlighting multiple strengths of our work. In particular, we appreciate the recognition of the clarity and quality of the writing, figures, and background presentation **(tCBu)**; the conceptual contribution of formulating energy-functional recovery via inverse optimization within the JKO scheme **(XRFW, vGio)**; the significance of our recovery guarantee in Theorem 3.1 **(XRFW, tCBu)**; our use of standard neural architectures such as MLPs instead of input-convex networks **(XRFW)**; and the broader potential of our reformulation of the JKO functional **(vGio)**.

To further improve the paper, we have reorganized and clarified the Appendix and made several targeted revisions (highlighted in orange in the updated manuscript) in direct response to the reviewers’ suggestions:

- **(tCBu)** Added clarifications in Section 3.2 regarding the interchange of the $\min$ and $\sum$.

- **(XRFW, tCBu)** Expanded the Related Works section in Appendix A.1 to include all suggested references.

- **(tCBu)** More explicitly emphasized in Section 4.2 how our approach improves over prior JKO-based methods [1, 2].

- **(XRFW, tCBu, vGio)** Added a full overview of all energy parametrizations (potential, interaction, internal) in Table 1, and clarified why high-dimensional experiments focus on potential energies.

- **(XRFW)** Added a detailed discussion in Appendix A.3 comparing paired and unpaired settings, supported by Figure 3 and an ablation study on the fraction of paired data, i.e. full trajectories available during training.

- **(XRFW, tCBu, vGio)** Included new paired experiments for learning interaction energies in Appendix B.1.2, showing that even in the paired case certain interactions cannot be reliably recovered (Figure 5), further reinforcing the challenges illustrated in the unpaired case (Figure 6).

- **(tCBu, vGio)** Added details in Appendix C.3 on the stability of our method, strategies considered for improving it, and remarks on scalability and efficiency of map computation.

We believe these revisions substantially strengthen the manuscript and address the reviewers’ concerns.

Thank you once again for the constructive feedback. Detailed responses to individual comments follow.

---
**References**

[1] Bunne, Charlotte, et al. "Proximal optimal transport modeling of population dynamics." International Conference on Artificial Intelligence and Statistics. PMLR, 2022.

[2] Terpin, Antonio, et al. "Learning diffusion at lightspeed." Advances in Neural Information Processing Systems 37 (2024): 6797-6832.

---

### Author Response · Authors · 2025-12-03
**Rebuttal Summary**

We once again sincerely thank all the reviewers for their valuable feedback. We believe we answered all the reviewer questions during the rebuttal and have provided a revision with additional clarifications.

The primary reviewers' concern involved our focus on **potential energy** in the main experiments. We have now **clarified this rationale directly** and added **supporting experiments in the main text** to reinforce our choice as well as additional clarification in the appendix.

We are encouraged that the two reviewers (**XRFW, tCBu**) who responded to our rebuttal both reacted positively, acknowledged the contribution, and **raised their scores**.

---

### Meta-Review · Area_Chair_ejL3 · 2026-01-06

**Summary:**

The paper contributes iJKOnet, an approach that combines the JKO framework with inverse optimization techniques to learn population dynamics. The authors cast the energy functional recovery as a min-max optimization problem and provides theoretical guarantees for narrower case of potential energy recovery.

The reviewers noted several concerns including the gap between the claimed generality suggesting arbitrary free-energy functional recovery and the demonstration of potential energy only and of the stability of adversarial training and computational scalability of the method. They also noted missing references.

The rebuttal and updated manuscript clarified that while the method indeed fails to solve the general energy functional problem, it nevertheless improves upon the state of the art of potential energy recovery in both theory and practice. The authors also added missing references.

The paper received split reviews with two borderline accepts and one strong reject. While acknowledging the concerns of over claiming, the paper still appears to improve upon the state of the art. Therefore I recommend weak accept.

**Reviewer Concerns:**

The reviewers noted several concerns including the gap between the claimed generality suggesting arbitrary free-energy functional recovery and the demonstration of potential energy only and of the stability of adversarial training and computational scalability of the method. They also noted missing references.

The rebuttal and updated manuscript clarified that while the method indeed fails to solve the general energy functional problem, it nevertheless improves upon the state of the art of potential energy recovery in both theory and practice. The authors also added missing references.

**Reviewer Scores:**

I suspect all reviewers would have raised their scores modestly.

---

### Decision · Program_Chairs · 2026-01-26

Accept (Poster)